# The crucial role of circular waste management systems in cutting waste leakage into aquatic environments

Adriana Gómez-Sanabria [1] ✉ & Florian Lindl [1]

Waste leakage has become a major global concern owing to the negative impacts on aquatic ecosystems and human health. We combine spatial analysis with the Shared Socioeconomic Pathways to project future waste leakage under current conditions and develop mitigation strategies up to 2040. Here we show that the majority (70%) of potential leakage of municipal solid waste into aquatic environments occurs in China, South Asia, Africa, and India. We show the need for the adoption of active mitigation strategies, in particular circular waste management systems, that could stop waste from entering the aquatic ecosystems in the first place. However, even in a scenario representing a sustainable world in which technical, social, and financial barriers are overcome and public awareness and participation to rapidly increase waste collection rates, reduce, reuse and recycling waste exist, it would be impossible to entirely eliminate waste leakage before 2030, failing to meet the waste-related Sustainable Development Goals.

The world is facing a critical waste disaster resulting from the rapid increase of waste generation and the inability to cope with it in a sustainable manner endangering the environment, climate, and human health[1]. Estimates show that future global municipal waste generation is expected to increase between 20% (sustainability pathway) and 68% (fossil-fueled pathway) by 2050 depending on the assumed socio-economic pathway[2]. The composition of waste is becoming increasingly complex and if waste treatment were to stagnate at current levels the negative consequences could be exacerbated further[3].

Currently, 64% of global municipal solid waste (MSW) generated is mismanaged, 29% of which is open burned, 18% ends up in dumpsites and 17% is scattered[2]. Scattered waste is dynamic, meaning that depending on its physical characteristics and certain climate and geographical conditions, it can be mobilized, damaging terrestrial and aquatic ecosystems[4]. Land-based waste has been identified as the main source of marine litter[3] (of which 80% is plastic waste[5]). Although initiatives to stop plastics from entering the oceans exist[5], without appropriate waste management systems it is impossible to stop leakage of waste into our ecosystems[3].

Current debates on marine litter focus mainly on plastic waste (macro- and microplastics) due to its toxicity to aquatic life and negative effects on human health[6]. Global efforts to combat plastic pollution include the amendments to the Basel Convention in 2019 with the aim to monitor transboundary movements of plastic waste[7] and more recently the Resolution to End plastic pollution by the United Nations Environment Assembly in 2022[8]. As of 2018, 127 countries have adopted some form of legislation to regulate plastic bags and single-use plastic items[9]. Furthermore, scientific research on marine litter at a global[3,4,10] and regional[11] levels concentrates mainly on plastic rather than on the underlying problematic related to waste management. Global estimates suggest that in 2010, 275 Mt (million tons) of plastic waste was generated across 192 coastal countries, of which between 1.75% and 4.61% ended up in the ocean[3]. In 2019, it was estimated that 1000 rivers are responsible for 80% of the annual global plastic emissions into the ocean with an average of 1.75 Mt per year[10]. A more recent study shows that litter is a global and heterogenous problem that requires sub-national approaches when adopting solutions[4]. The same study identifies that some of the most polluted sites are located in places with high-infrastructure but low-wealth such

[1]Pollution Management Research Group. Energy, Climate and Environment Program, International Institute for Applied Systems Analysis, Laxenburg, Austria. ✉e-mail: gomezsa@iiasa.ac.at

as the cities of Athens, Tunis and Lima[4]. Regional studies include an analysis of rural plastic emissions into the Izvoru Muntelui lake (Eastern Carpathians) which suggests that rural municipalities might be responsible for 85.51% of plastic bottles collected between 2005 and 2010 and it concludes that plastic pollution is mainly local[11]. A more recent study on the Carpathian region identifies that watercourses below 750 m.a.s.l are significantly affected by mismanaged plastic waste and most of the hotspots are located in Romania, Hungary, and the Ukraine[12]. Another study revealed that 24.3% of waste generation in Jakarta and Bandung (Indonesia) ends up in waterways with highest plastic accumulation in the mainstream of the Ciliwung and Cikapundung rivers[13]. Moreover, a recent assessment demonstrates that United States generated the largest amount of plastic waste in 2016, of which between 0.14 and 0.41 Mt was illegally dumped and 0.15−0.99 Mt was exported as recycling material that ended up being inappropriate managed[14].

Furthermore, initiatives to reduce plastic waste leakages include the work and economic impact analyses of plastic pollution carried out by the International Union for Conservation of Nature (IUCN) in Fiji[15], Samoa[16], Vanuatu[17], Antigua and Barbuda[18], the Mediterranean islands[19], among others. It is important to note that the scope and methodologies to estimate plastic waste leakage differ. While some methodologies include macroplastics from production to use and fate[20] or are based on population and spatial analysis[21], others assess micro−and macro plastic waste leakage over the entire life cycle of a product (corporate plastic footprint)[22].

However, global studies that comprehensively analyze scenarios on how the improvement of waste management systems under future plausible pathways can reduce leakage of waste in terrestrial and aquatic environments are rather limited. To our knowledge, no global assessment exists that combines the Shared Socio-economic Pathways (SSPs)[23], waste generation and management storylines and spatial analysis of urban and rural areas to project future waste leakage and analyze the mitigation potential of circular waste management systems to cut leakage of waste into aquatic environments (lakes, rivers and coastal areas).

For this research, we combine our more recent method to globally assess the current and future MSW generation and composition in urban and rural areas[2] with spatial analysis to identify potential global MSW leakage hotspots in aquatic environments and potential reduction strategies. We distinguish between rural and urban areas under five future socioeconomic pathways up to 2040. Each of the scenarios include a "Baseline" and a "Maximum Technically Feasible Reductions" scenario. The "Baseline" includes waste-related legislation adopted until 2018. Our detailed representation goes beyond the estimation of plastic waste into aquatic environments but rather attempts to quantify the MSW leakage as a whole (including eight different waste streams). The differentiation of urban and rural areas in our spatial analysis for MSW generation reflects the disparities of lifestyles, income, and resource consumption within a country/region[24]. This allows us to analyze the MSW leakage problem from a holistic waste management systems perspective. The IIASA-GAINS model is used as a framework to carry out this assessment. The GAINS model has global coverage with a geographic representation of 180 countries/regions with multitemporal resolution at 5-year intervals. The MSW sector in the model further differentiates between urban and rural areas within a country/region.

The results of this study can be further developed in combination with additional environmental, meteorological, and geographical variables, as demonstrated in ref. 10, who included characteristics such as slope, precipitation, stream order, and river discharge to estimate the amounts of MSW potentially reaching the oceans as well as their estimated origin. The outcomes of this study can also serve as science-based evidence to support the development of the new treaty to move towards a legally binding instrument to end plastic pollution[8] and to help establish a global standardized MSW reporting framework.

## Results

### Scenarios of scattered municipal solid waste up to 2040

MSW generation and composition is estimated by using different elasticities representing four different income averages assuming that MSW composition is dependent on average national income levels as stated in ref. 2. Global MSW generation is estimated at about 2560 Mt (million ton) in 2020[2] and it is expected to increase to 3320−3790 Mt in 2040 depending on the followed socio-economic pathway[2] (Fig. 1). The estimates show that in 2020 the world generated 1091 Mt of food waste (43% of MSW), 260 Mt of plastic waste (10%), 366 Mt of paper waste (14%), 113 Mt of glass waste (4%), 73 (3%) Mt of metal waste and 651 Mt (26%) of other waste (including, textile, wood, and mixed waste). The results by stream are in line with those assessed by ref. 25 and with ref. 21 for plastic waste (239 Mt)[21] in 2020. Future average global composition of MSW will see a slight decline of organic waste fraction (food) in all SSPs, except in the SSP3. Nonetheless, food waste will remain as the highest portion of MSW in the future across all SSPs in absolute terms, finding that is in agreement with ref. 25, In 2040, food waste is expected to increase between 26% and 40%, plastic waste between 37% and 45% and paper waste between 27% and 50% depending on the socio-economic pathway when compared to 2020 quantities. When looking at the composition of MSW the SSP1_MFR in 2040 stands out with the share of food waste being reduced to 28% due to implementation of the food waste reduction target. Future paper waste fraction may increase due to the reduction of plastic under the assumption that the policy reducing plastic waste is adopted. Our results show that other waste streams can increase as a potential rebound effect of this measure (e.g., paper cups replacing single-use plastic cups). Therefore, targets to reduce other waste streams are also urgently needed.

The amount of scattered MSW, defined here as the fraction of MSW not collected and left over an area, is assessed for each stream (i.e., food, plastic, paper, etc.) by subtracting the amount of MSW open burned from the uncollected MSW (see Methods). Our results suggest that scattered MSW accounts for about 14% or 350 Mt of the total global MSW generation in 2020. This estimate assumes that part of the uncollected food waste in rural areas in developing countries is used as animal feed[26] or composted at household level[27] and therefore is not accounted for as scattered waste. Over 87% of the total scattered MSW is generated across China (30%), South Asia (20%), Africa (20%) and India (17%). Wealthier countries are associated with lower levels of mismanaged MSW[4] and hence their contribution to total scattered MSW is comparatively lower. The composition of scattered MSW in 2020 is assumed to be 52% (181 Mt) food, 9% (33 Mt) plastic, 8% (27 Mt) paper, 18 Mt (5%) glass, 11 Mt (3%) metal and 78 Mt (23%) other waste. The resulting composition of scattered waste is therefore highly depending on the levels of collection rates as well as type of management by stream. In regions such as Africa, South Asia, China, and Latin America and the Caribbean (LCAM), food waste accounts for the highest fraction of scattered waste (between 49% and 53% of the total MSW) while in regions like EU27 + UK and Russia, plastic and glass waste make up the majority of scattered waste. The proportion of paper waste in North America and Oceania OECD and mixed waste in India in the scattered waste fraction is substantial (see Supplementary Information Fig S1 and Fig S2). Our estimates show that urban areas currently account for about 70% of the total scattered MSW while rural areas account for the remaining 30% (Fig. 1). Scattered MSW is expected to increase to 427−475 Mt by 2040 if global MSW treatment were to stagnate at current level. The largest increase of scattered MSW in the future with the current management is expected in the SSP5 as this pathway exemplifies a world with high consumption patterns and high urbanization rates resulting in huge MSW generation quantities. Towards 2040, the contribution of rural areas decreases to 20% resulting from the migration of people to urban settings and continued trend of urbanization of rural areas, with an exception of the

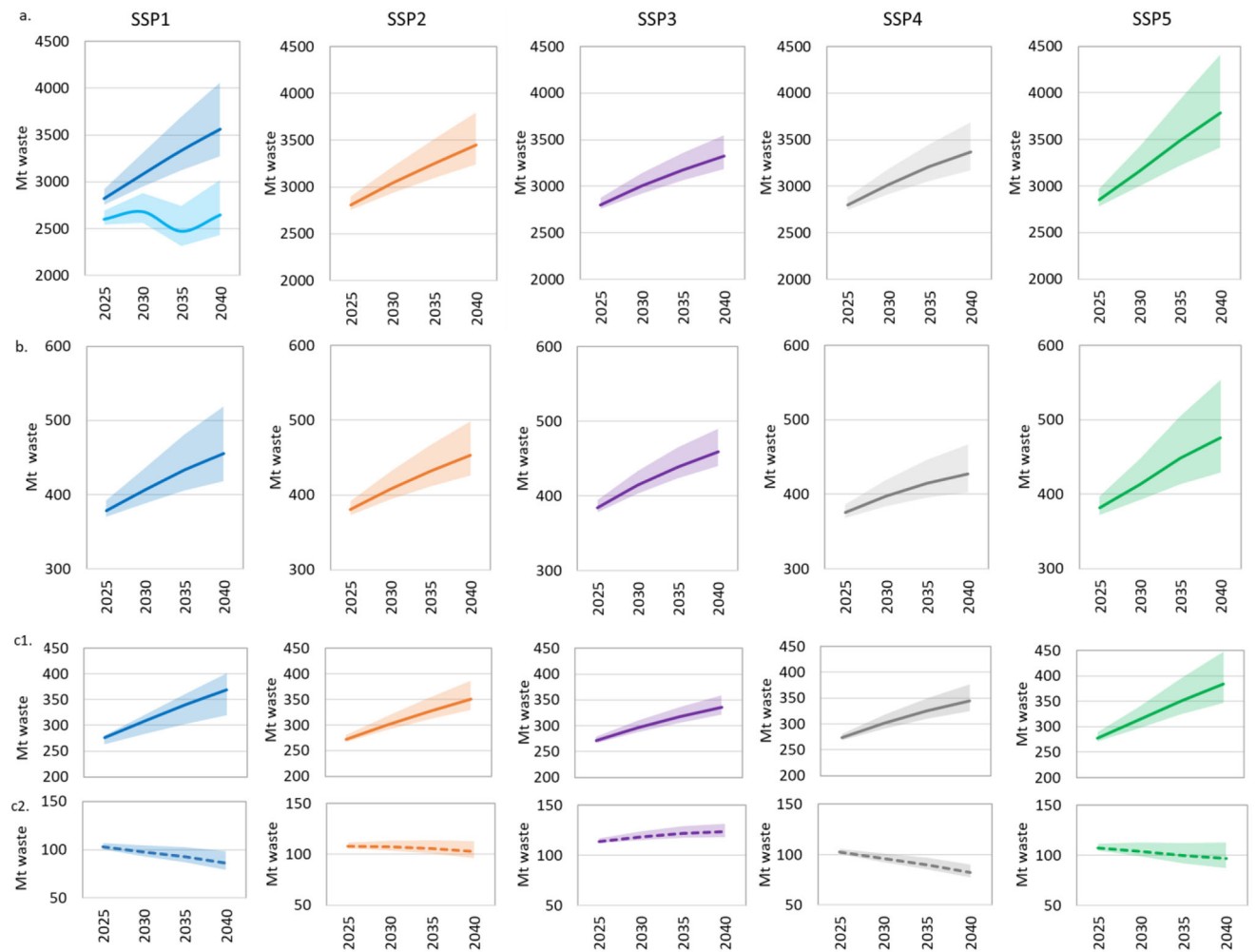

**Fig. 1 | Municipal solid waste (MSW) generation and scattered waste in million tons (Mt). a** Global total MSW generation, for SSP1 lighter blue represents the SSP1_MFR scenario, **b** Global total scattered MSW, **c** Scattered MSW in urban (**c1**) and rural areas (**c2**). Shaded area represents the interquartile range (IQR) from the 25th to the 75th percentile. The line within the shaded area represents the median value. Estimates by GAINS/country region are presented in the Supplementary Data 1. Table S1 Supplementary Information. shows the underlying data. CI for the model is presented in Table S6.

SSP3 in which population growth is high but urbanization is particularly slow, especially in low income regions[28]. Fig S3. shows scattered MSW by region in the *Baseline* scenario.

## Identification of MSW leakage in aquatic ecosystems and mitigation scenarios

Identification of potential MSW reaching rivers, lakes and coastal areas is based on the amount of uncollected scattered MSW by waste stream generated by the population residing at a distance up to 1 km from the aquatic systems. We divide the distance (0–1 km) into four buffer zones of 250 m each. We then apply different factors for each buffer zone representing the fate of MSW leakage according to the distance from rivers, lakes, and coastal areas distinguishing between urban and rural areas (see Methods). Note that we apply the same fate factors for all streams, however, we are aware that different streams may behave different according to the size, volume, density, degradability and may also be influenced by climatic and geographical conditions in a different way.

We modeled the baseline scenarios based on the assumption that countries implement the current MSW management legislation (up to 2018). We estimate that the global potential amount of MSW reaching rivers in 2020 is 74 Mt which is about 21% of the global estimated scattered MSW and 3% of the global MSW generation in 2020. China, South Asia, Africa, LCAM, and India account for 80% of MSW at risk of

reaching rivers. This shows the urgency of increasing MSW collection and improving waste management systems in these regions. Our estimations suggest that at global level, 70% of the MSW reaching rivers is occurring in urban areas with low collection rates whereas the remaining 30% happens in rural areas. This is consistent with ref. 10, which states that small rivers in urban areas are the most polluted. However, the distribution changes at regional level. For example, urban areas in China, South Asia, and India account for 70–80% of the MSW that can potentially reach rivers while in Russia and the Former Soviet Union rural areas are responsible for 70% of the possible MSW ending up in rivers. In this study, we found that potential food waste leakage into rivers amounts to 40 Mt in 2020. While the focus of current research on marine litter is mainly on plastic, adverse effects of food waste on ecosystems should not be neglected. Although food waste itself is biodegradable and non-hazardous, it certainly affects the food chain in aquatic ecosystems and can indirectly cause eutrophication and subsequent loss of biodiversity[29]. Additionally, we estimate that the potential global plastic waste leakage into rivers is 7.4 Mt in 2020. Estimates of plastic waste from rivers into oceans range from 0.41 to 4 Mt per year[10,30,31]. It is important to highlight that our study does not examine the transport of MSW in aquatic environments, but rather focuses on identifying the probable leakage locations. Potential MSW reaching rivers under the current waste management regimes is projected to increase by 30% compared to the baseline to a total of

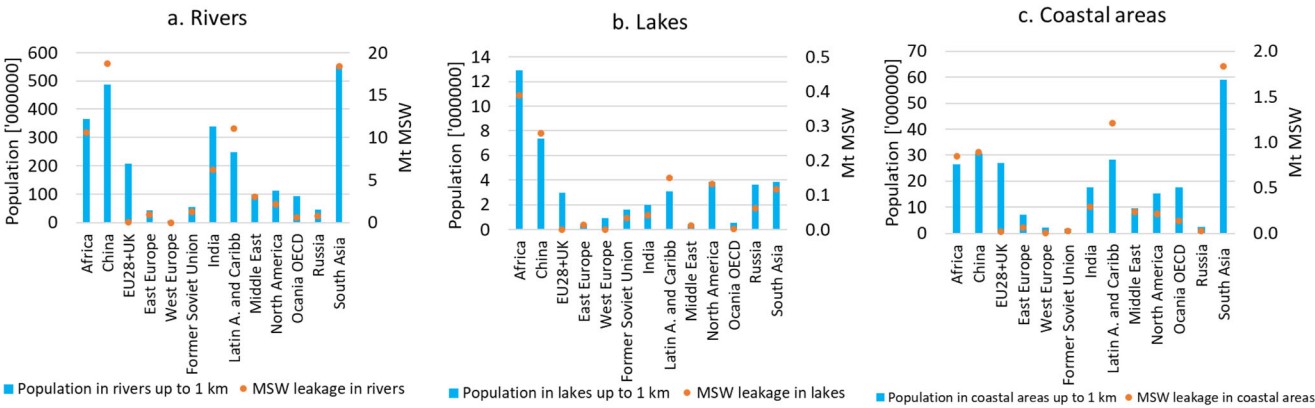

**Fig. 2 | Population living up to 1 km from aquatic environments and municipal solid waste (MSW) leakage in 2020. a** Rivers ($n = 153345$), **b** Lakes ($n = 3400$) and **c** Coastal lines ($n = 122$). Regions with high quantities of mismanaged MSW and highly populated areas close to aquatic ecosystems have the greatest likelihood to become important MSW leakage hotspots.

90 Mt (SSP4) and 100 Mt (SSP5) per year by 2040. EU27 + UK and Oceania OECD will see a reduction of MSW leakage in rivers (and in all other aquatic ecosystems) already in the *Baseline Scenario* resulting from the implementation of policies and strategies tackling improvements in waste management rather than in reduction of waste generation. These policies include the EU Waste Framework Directive 2008/98/EC[32] and the amendment EU Directive 2018/851[33], the EU Directive on Packaging and packaging waste 94/62/EC[34] and the amendment EU Directive 2018/852[35], the EU Directive 2019/904[36], the EU New Circular Economy Action Plan (which includes the European Strategy for Plastics)[37], the Circular Plastic Alliance[38], the European Plastic Pact[39], the 3 R's strategy in Japan[40] and the National Waste Policy Action Plan 2019 in Australia[41].

Furthermore, our assessment indicates that 1.35 Mt of MSW potentially entered lakes in 2020. Around 58% of MSW enters the lakes in urban areas and 42% in rural areas. Our results suggest that Africa and China are responsible for 55% of the assessed total MSW entering lakes. This is related to the fact that these two regions have the highest population residing in close vicinity (up to 1 km) to lakes among all regions and at the same time exhibit high quantities of scattered MSW resulting from the lack of MSW management systems. Although India, South Asia, and LCAM have significant quantities of scattered MSW, population around lakes is significantly lower compared to China and Africa, and therefore the likelihood of leakage of MSW into lakes in these regions is reduced. Apart from the potential food waste ending up in lakes (0.73 Mt), plastic (0.13 Mt) and paper (0.11 Mt) account for a large portion of the probable MSW entering lakes. To our knowledge, studies assessing global leakage of waste entering lakes are not available and therefore a comparison of our results to other studies is not possible at the current stage.

Our results further suggest that 5.79 Mt of scattered MSW near coastlines might have entered seas in 2020, of which 3.01 Mt is food waste, 0.57 Mt is plastic waste, and 0.52 Mt is paper waste. We do not compare our estimate of the total amount of MSW entering seas from coastal countries to other studies, instead we present a comparison at plastic waste level. In that context, ref. 3 estimates that between 4.8 and 12.7 Mt of plastic waste from 192 coastal countries entered the ocean in 2010. Our estimate is six orders of magnitude lower than the average estimated by ref. 3 One of the reasons of this difference apart from the methodology is that our study considers people residing within 1 km of the coastline while ref. 3 considers population living within 50 km and does not distinguish between urban and rural settings. This clarifies again that population is a major driver of MSW generation and highlights the added value of spatially explicit analysis differentiating between urban and rural population. Our results show that most of the scattered MSW in coastal areas hypothetically comes

from South Asia and LCAM, contributing around 50% of the total. South Asia is the region with the highest population residing withing 1 km of coastal areas and therefore special attention to reduce leakage in coastal areas should be paid in those regions. If current trends persist at global level, leakage of MSW in coastal areas is expected to increase by 29% and 44% in the SSP3 and SSP5, respectively.

Overall, our study estimates that 80.8 Mt of MSW leaked into aquatic environments in 2020 and is projected to increase by a maximum of 36% in 2040 driven by increases of MSW generation under the *Baseline* scenarios (Fig. 2., presents population living up to 1 km from aquatic environments and leakage of MSW, and Fig. 3., shows the top ten highest MSW leakage hotspots for each category -rivers, lakes, and coastal areas- in 2020 and Fig S3 shows the top 20 countries with the highest estimated leakage of MSW into aquatic environments).

Leakage into rivers accounts for 91% of the total MSW reaching aquatic systems and most of the leakage occurs in urban settings. This finding is somehow in accordance with that in ref. 13 who found urban areas in Jakarta and Bandung emitted the highest waste into the water way. (Fig. 4., displays a map of potential MSW leakage in rivers in 2020). This is mainly attributed to the fact that most of the population is concentrated in urban areas as identified in ref. 42 for plastic waste. Our results indicate that 8.09 Mt of plastic waste reached aquatic environments in 2020. This estimate is in line with the 6.1 Mt in 2019 presented in the Global Plastics Outlook (2022)[43]. However, our result is around 60% lower compared to the average estimate (19–23 Mt) presented in ref. 44 in 2016 (Table S2 shows a collection of studies estimating waste (mainly plastic) leakage into aquatic environments). These huge differences in the estimates highlight the complexity of measuring not just plastic waste leakage, but in general MSW flows. Therefore, actions need to be taken to develop a standardized reporting framework that can support the monitoring of MSW generation, composition, and flows, and follow up the implementation of actions (including political, economic, and technological measures) targeted to the reduction of MSW and improvement of waste management systems. A standardized framework will reduce the uncertainty of the assessments and will provide better knowledge and information to develop strategies to tackle the MSW crises. This framework can also contribute to monitor the progress of the circular economy regarding availability and flows of secondary materials.

Our mitigation scenarios adopt the socio-economic narratives from the Shared Socioeconomic Pathways (SSPs) (see ref. 28). The SSPs provide five plausible pathways about the world's probable socioeconomic development. Based on our interpretation of the narratives, we develop circular MSW management scenarios representing mitigation and/or adaptation challenges. The implementation of circular waste management systems are developed in accordance with

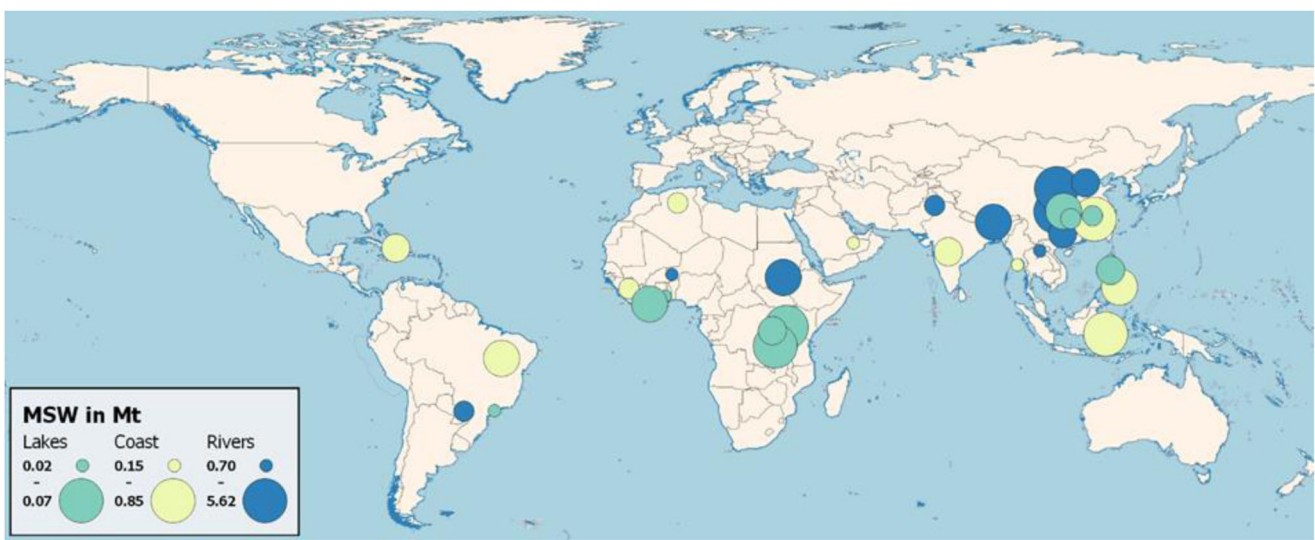

**Fig. 3 | Overview of the ten highest potential municipal solid waste (MSW) leakage by category (rivers, lakes, and coastal areas), sized relative within their category.** Tables by category are presented in the Supplementary Information S4. HydroRivers data as of October 04, 2022, HydroLakes data as of October 27, 2022, FAO Administrative Boundaries and Coastlines data as of October 04, 2022. Created using QGIS 3.26.1 (https://www.qgis.org/en/site/).

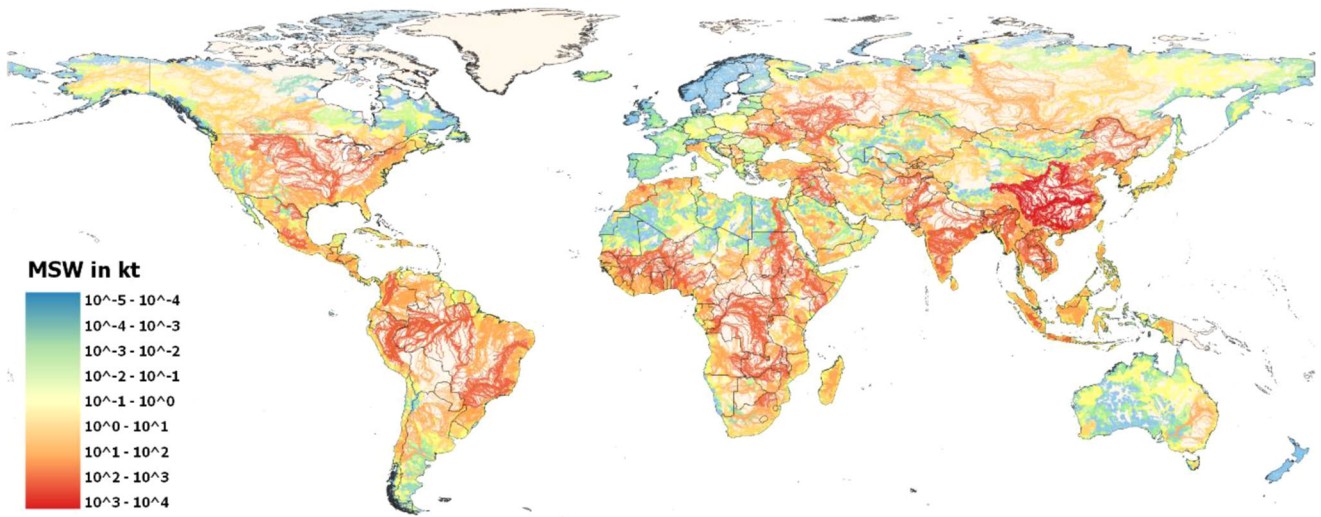

**Fig. 4 | Map of potential municipal solid waste (MSW) leakage in rivers in 2020.** In the GAINS model, income per capita is used as a driver to estimate MSW generation. Waste management and adoption of policies until 2018 are adopted in the model. HydroRivers data as of October 04, 2022, FAO Administrative Boundaries and Coastlines data as of October 04, 2022. Created using QGIS 3.26.1 (https://www.qgis.org/en/site/).

the EU's waste management hierarchy—Directive 2008/98/EC [32] as stated in ref. 2. A circular waste management system is defined here as a system with successful implementation of MSW reduction policies by reducing food and plastic waste generation, maximum technically feasible recycling rates of all MSW streams, and once recycling capacity is exhausted, incineration of refuse MSW with energy recovery. Furthermore, anaerobic digestion is implemented to treat food and garden waste, high diversion of MSW from landfills and upgrading of dumpsites [45]. A description of the narratives in terms of economic development and demographics for each SSPs along with the description of MSW management scenarios is presented in the supplement S6. Note that mitigation scenarios are represented by an additional MFR in the naming (e.g., SSP1_MFR).

Mitigation scenarios show that tackling the reduction of food and plastic waste as a behavioral measure together with the global improvement and adoption of circular MSW management systems in the SSP1_MFR results in earlier decline of global scattered MSW and therefore faster reduction of leakage into aquatic environments in both, urban and rural areas. In the SSP1_MFR, scattered MSW is expected to decrease by 50% in 2025, 73% in 2030 and close to 99% in 2040 compared to the corresponding baselines, as a result of increases in collection rates, material recycling and anaerobic digestion. 35% and 45% of total MSW generation will be recycled in 2030 and 2040, respectively. 55% and 98% of the total food waste generated will be treated in anaerobic digestion with biogas recovery. Towards 2040, the world will divert 98% of the MSW from landfills. This means, however, that even the sustainability scenario (SSP1) [28] will not be enough to achieve the UN 2030 Agenda [46] goal to: By 2025, prevent and significantly reduce marine pollution of all kinds, in particular from land-based activities including marine debris and nutrient pollution. This also implies that at global level more effort is needed to fully adopt circular MSW systems by 2030, thereby increasing the proportion of population with access to MSW collection services, which will require faster and wider development of physical infrastructure (e.g., road networks), and improving MSW management in high-tech facilities (SDGS indicator 11.6.1) [46]. The results of this scenario demonstrate

### a. SSP1_Baseline in 2030

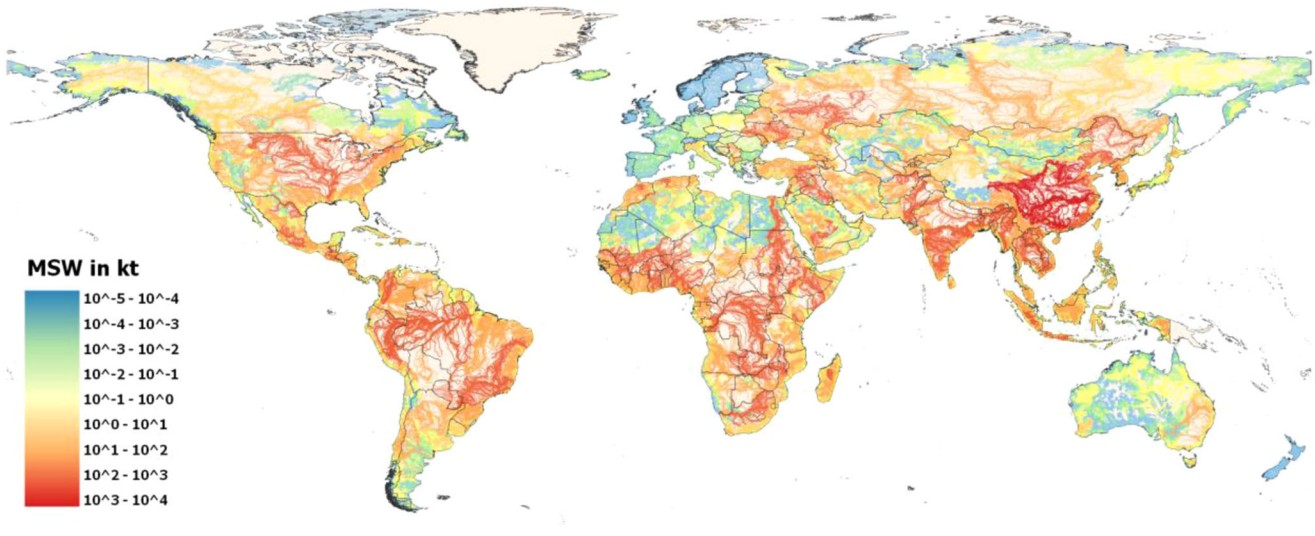

### b. SSP1_MFR in 2030

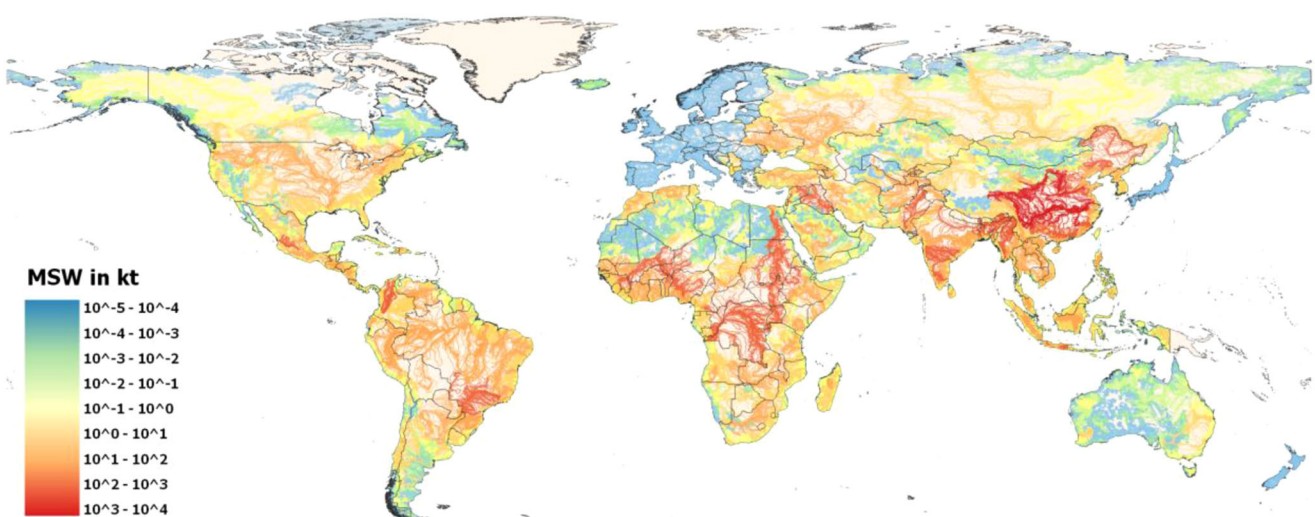

**Fig. 5 | Map of potential municipal solid waste (MSW) leakage in rivers in 2030 in the SSP1. a** SSP1 Baseline Scenario, **b** SSP1 Mitigation Scenario. A notable reduction of MSW leakage is observed in LCAM, East Europe and some countries in Africa. Even though leakage in rivers decreases in China, India, and South Asia by 2030, faster interventions are required to significantly bring down MSW leakage in areas located around the most polluted rivers. HydroRivers data as of October 04, 2022, FAO Administrative Boundaries and Coastlines data as of October 04, 2022. Created using QGIS 3.26.1 (https://www.qgis.org/en/site/).

that reduction of scattered waste can only be solved by implementing integrated strategies in the MSW sector rather than solely focusing on one stream (e.g., plastics) or one strategy (e.g., recycling).

At regional level, SSP1_MFR shows that China, South Asia, Africa, India and LCAM will benefit the most from an improved cooperation at local, national, and international level to implement circular MSW management systems. The main obstacles in these regions such as lack of regulations and implementation, low collection rates, high transportation costs and lack of diversified MSW technologies[47] are partly overcome through adequate human and financial resources[28]. These regions account for 88% of the total global reduction of scattered MSW in 2030 compared to the baseline. However, 128 Mt of MSW will be still released to the environment in 2030. By 2040, it is expected that under the global fulfillment of the objectives of this scenario, scattered

MSW will be virtually eliminated (>99%). This translates to a reduction of MSW leakage into aquatic environments of 68–75% by 2030 and could almost entirely eliminate MSW leakage by 2040 compared to their baseline. In other words, the leakage of 73 Mt in 2030 and 110 Mt of MSW in 2040 into aquatic environments will be avoided in the SSP1_MFR as a result of the proactive environmental management. Figure 5 displays a map comparing the leakage of MSW in rivers in SSP1 Baseline and SSP1_MFR in 2030.

Likewise, reduction of MSW leakage in the SSP5_MFR follows a similar trajectory as the SSP1_MFR in both, urban and rural areas. Although technological development in the SSP5_MFR is high, the world focuses on economic growth rather than on reducing MSW generation. In this scenario most of the population will live in urban areas. This is accompanied by excessive resource use and

consumption patterns resulting in high MSW generation amounts. While in the SSP1_MFR environmental aspects are essential, in the SSP5_MFR a lack of environmental concerns results in implementation of *only* end-of-pipe solutions without considering behavioral or any additional aspects (i.e., reduction of MSW generation). Both scenarios will virtually eliminate leakage of MSW in 2040. However, the SSP1_MFR will bring faster reductions in MSW leakage due to the combination of technological development with high environmental awareness. This finding highlights the significance of adopting behavioral measures (i.e., reduction of food and plastic waste) in addition to technical and institutional arrangements to cut waste leakage. A timely implementation is necessary to avoid the accumulation of MSW in both, land and aquatic environments.

In the SSP2_MFR, the adoption of circular MSW management systems will happen at slower pace compared to the SSP1_MFR or SSP5_MFR, especially in rural areas, as a result of the moderate economic development and some inequalities in developing countries. Therefore, leakage reduction strategies in the SSP2_MFR will not bring major progress before 2035. In this Middle of the Road scenario we see regions with adequate existing MSW management systems such as Europe and Oceania OECD keep advancing them and further develop strategies to reduce MSW generation. However, the adoption of these strategies is still uncertain. In this scenario ~35 Mt of MSW are still at high risk to end up in aquatic environments in 2040. 95% of the expected MSW leakage in aquatic environments in 2040 will potentially happen in South Asia, China, Africa, LCAM, and India. China and South Asia alone will account for around 55% of the total leakage.

The SSP3_MFR depicts a fragmented world in which population growth is high, but urbanization slow. MSW generation is lower compared to the other SSPS (except SSP1_MFR), however, the low economic development and lack of international cooperation hinders the fast and fully adoption of circular MSW management systems before 2040. Developing countries, and especially their rural areas, are the most negatively affected by the slow economic growth and technological disparities[28]. In 2040, roughly 40 Mt of MSW are at risk of finding their way into aquatic environments. As expected, the regions more negatively affected will be South Asia, China, Africa, LCAM, and India in which collection rates will be lower in rural areas compared to urban areas. Therefore, MSW management in these regions will see only a very small reduction of MSW leakage into aquatic environments before 2035, with only developed regions such as Northern America and Europe seeing comparable reductions to other mitigation scenarios (Fig. 6). India and China will even face an increase of leakage into rivers due to the growth of rural population living close (up to 1 km) to water courses in 2025.

Our results for the SSP4_MFR scenario are somehow in between SSP2_MFR and SSP3_MFR. This scenario describes a world with moderate population and economic growth. Improvement of MSW management systems will be faster in urban areas than in rural areas, particularly for developing countries. Similar to the SSP3_MFR, MSW collection rates in rural areas will lag behind the urban areas. Developing regions face similar issues as in the SSP3_MFR and therefore leakage reduction is slow, yet not as slow as in the SSP3_MFR. Scattered MSW will be reduce to ~14 % in rural and 4% in urban areas in 2040.

Figure 7, shows the number of people residing up to 1 km from rivers, lakes, and coastal areas and global MSW leakage into aquatic environments for *Baseline* and *Mitigation* scenarios(results by scenario, by region, and by country are presented in the Supplementary Data 1). In addition, the development of MSW management by mitigation scenario is presented in the Supplementary Information S8.

In summary, here we present a detailed assessment of the crucial role the adoption of circular MSW management systems plays in effectively cutting leakage of MSW in aquatic environments. Our study indicates that leakage of MSW into aquatic environments depends mostly on the level of waste collection. In addition, MSW leakage also

dependents on population size, populated area (urban-rural), physical environment, MSW generation and composition, and level of MSW management systems. Results that expand the findings of ref. 3 and ref. 21 for plastic waste. The heterogeneity of conditions between and within countries results in different amounts of mismanaged MSW and contributions to MSW leakage into aquatic environments. Not in all cases countries with the largest amount of mismanaged MSW are the same countries with the highest quantities of MSW leakage into aquatic environments. Furthermore, current initiatives to stop MSW, especially plastics, from entering the ocean by stopping them at the river mouth or removing them from oceans form important actions but represent strictly passive adaptation. Our study demonstrates that circular MSW management systems could rather take a central role in active mitigation and could stop, not just plastics, but in all types of MSW (e.g., glass, metal, textiles, paper, food, etc.) from entering the ecosystems in the first place.

Future MSW generation is expected to increase according to the underlying socio-economic characteristics of the SSPs. Measures targeting the reduction of MSW generation will play an important role in decoupling MSW generation from GDP growth. Our results demonstrate that the reduction of scattered MSW can only be achieved by implementing integrated strategies in a holistic way in the MSW sector rather than solely focusing on one stream (e.g., plastics) or one strategy (e.g., recycling), thereby avoiding potential rebound effects of measures targeting specific streams (e.g., increase of paper cups replacing single-use plastic cups).

Leakage into rivers accounts for 91% of the total MSW reaching aquatic systems and most of the leakage occurs in urban settings. This means that if future MSW management is maintained at the current level and urbanization increases then scattered MSW will rise thereby increasing the quantities of MSW leakage at risk of ending up into aquatic environments. China, South Asia, Africa, and India account for the majority (~70%) of the potential leakage of MSW into aquatic environments. China is the region with the highest population residing within 1 km of rivers, South Asia is the region with the highest population residing withing 1 km of coastal areas, and Africa with the highest population living close to lakes (up to 1 km). It is crucial to act accordingly in these regions to diminish the future potential MSW leakage associated with the lack of management systems. China and India have also been found as target regions to diminish leakage of plastic waste in the environment[21].

By contrasting baseline and mitigation scenarios, our study indicates that in a future in which the world is assumed to be fragmented (SSP3_MFR) and unequal (SSP4_MFR), with weak institutions, lack of financial resources, the absence of knowledge and obstructed access to technology make the provision of circular MSW management systems difficult. As a result, projected MSW generation growth would exceed the efforts to mitigate MSW leakage. MSW management in rural areas in developing countries will lag behind urban areas. South Asia, China, Africa, LCAM, and India will be the most affected regions. On the contrary, in a world in which these barriers are overcome and policies, public awareness and participation to reduce, reuse and recycling MSW exists, it would be possible to mitigate and virtually eliminate MSW leakage on land and in aquatic environments. This will require scaling up separate collection, anaerobic digestion, and recycling capacities as well as diversion of MSW from landfills and upgrading of dumpsites. The effort to increase MSW collection and recycling will require the fully and fair integration of waste pickers who play a central role in the low and -middle income countries[48]. Our results shows that even in a best case scenario representing a future sustainable world (SSP1_MFR), the waste-related SDGs will not be met, highlighting the urgent need for additional efforts on strengthening MSW reduction strategies if these targets are to be met.

Some of the main challenges to develop strategies to mitigate MSW leakage in both, land, and aquatic environments, include the lack

### a. SSP3_Baseline in 2030

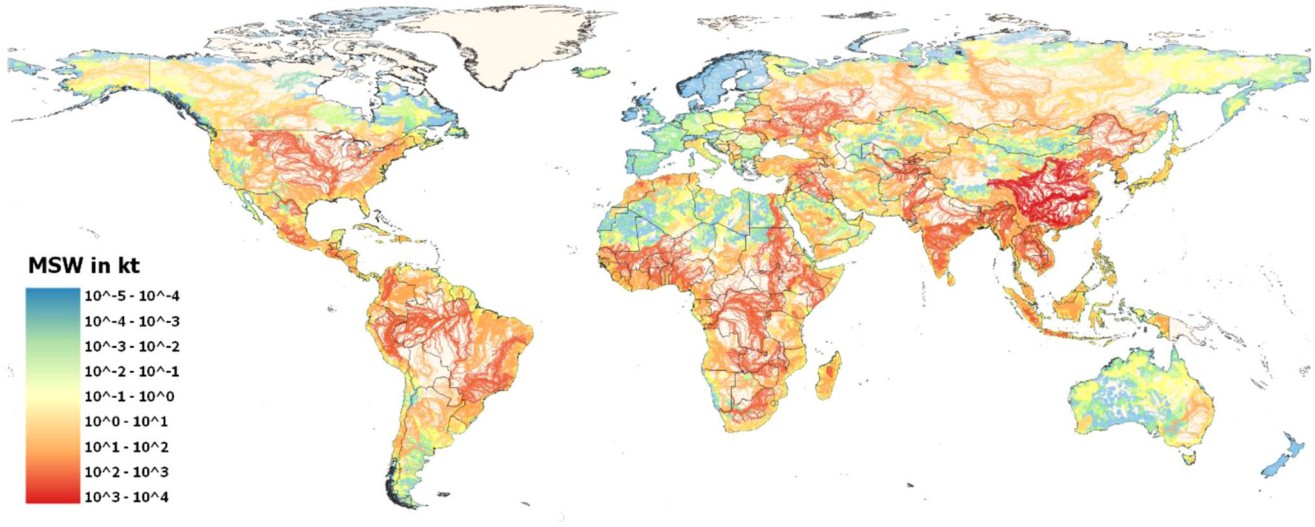

### b. SSP3_MFR in 2030

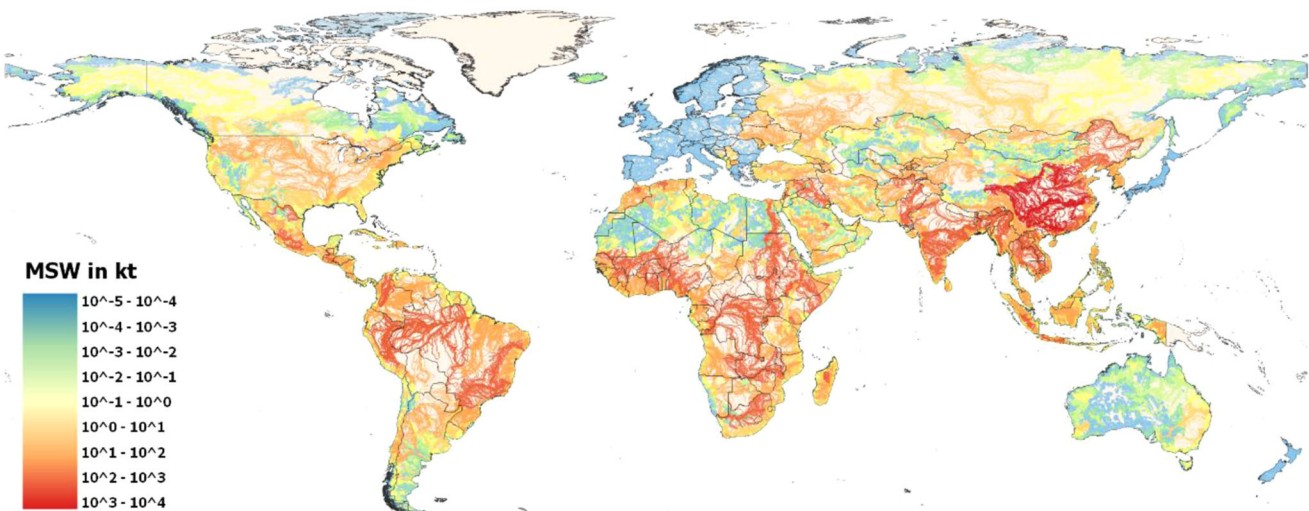

**Fig. 6 | Map of potential municipal solid waste (MSW) leakage in rivers in 2030 in the SSP3. a** SSP3 Baseline Scenario, **b** SSP3 Mitigation Scenario. Adoption of circular MSW management systems is slow and therefore reduction of MSW, especially in the Global South is minimal. HydroRivers data as of October 04, 2022, FAO Administrative Boundaries and Coastlines data as of October 04, 2022. Created using QGIS 3.26.1 (https://www.qgis.org/en/site/).

of reliable information related to MSW generation and composition and the variety of methodological approaches which result in a wide range of estimates and high uncertainty. These challenges make the results of the different studies difficult to interpret by policy makers. As for the plastic mitigation strategy[42], MSW mitigation strategies require accountable metrics. Therefore, actions need to be taken to develop a standardized reporting framework that can support the monitoring of MSW generation, composition, and flows, and follow up the implementation of actions (including political, economic, and technological measures) targeted to the reduction of MSW and improvement of waste management systems. A standardized framework will reduce the uncertainty of the assessments and will provide better knowledge and information to develop strategies and take actions at different levels (i.e., regional, urban-rural, streams) to tackle the MSW crises. This framework can also contribute to monitor the progress of the circular economy regarding availability and flows of secondary materials.

## Methods

The Greenhouse Gas-Air pollution interactions and Synergies (GAINS)[49] (http://gains.iiasa.ac.at/) model from the International Institute of Applied Systems Analysis is used to carry out this assessment, especially the MSW module[2,45,50]. The GAINS model is an integrated assessment model that provides an authoritative framework for assessing strategies that reduce emissions of multiple air pollutants and GHG at least cost, and minimize their negative effects on human health, ecosystems, and climate change. The GAINS model has global coverage with a geographic representation of 180 country/regions (Table S7 lists the countries/regions in GAINS) with multitemporal resolution at 5 years intervals. The methodology to estimate potential leakage of MSW and identification of hotspots involves the following steps:

### Model parametrization

The MSW module in GAINS integrates socio-economic variables which are exogenous to the model (i.e., population, urbanization,

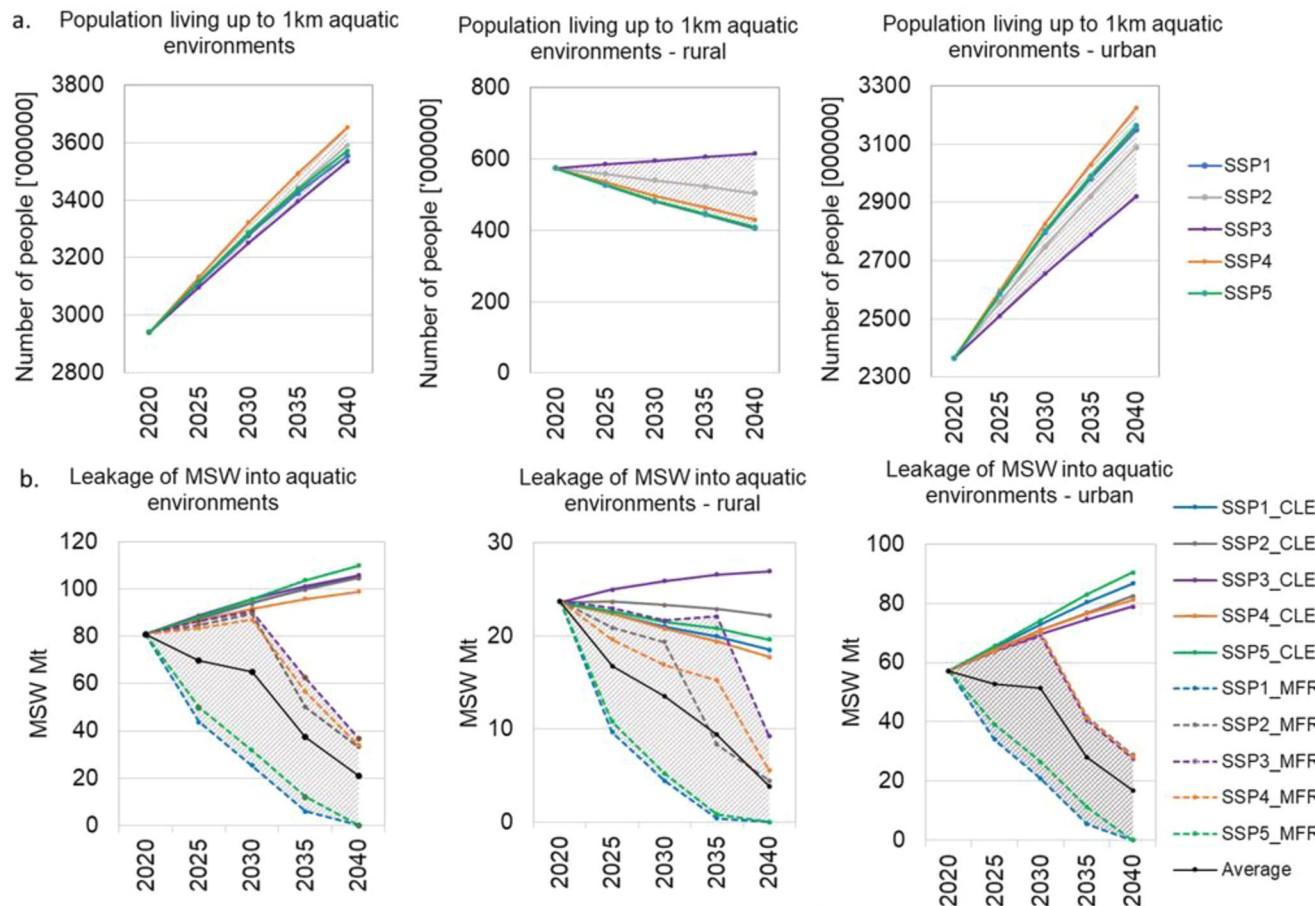

**Fig. 7 | Population residing close to aquatic environments and municipal solid waste (MSW) leakage. a** Population living up to 1 km around aquatic environments (rivers, lakes, coastal areas) by SSP, **b** Global leakage of MSW into aquatic environments. Dashed lines represent mitigation scenarios and shaded area shows the mitigation range.

GDP per capita), waste generation quantities, waste composition, waste control measures (i.e., waste treatment) and emission factors to quantify waste flows and to estimate air pollutants and GHG emissions. The MSW module in the model differentiates between urban and rural areas within a country/region and types of waste management by stream. This allows to capture national and regional disparities to a certain extent. The model links waste generation and waste composition to the type of treatment by applying the matrix presented in Table 1. The model constrains the treatments when not applicable to a specific waste fraction. The applicability of the waste control measures by type of waste should always sum up to 100%.

The model allows to represent actions and political interventions such as reduction of waste by intervening the activity data (e.g., waste generation and composition), represent variations/improvements in waste management systems by modifying the applicability rate of the waste control measures, and quantify the circularity of the waste systems by indicators such as landfill diversion rates, recycling rates by stream, anaerobic digestion and composting rates, carbon flows and energy generation. By contrasting baseline with mitigation scenarios, the model provides an analytical information to support the analysis of environmental, economic, and social impacts. In this context, the MSW module in GAINS offers the opportunity to analyze the co-benefits of adopting circular waste management systems on GHG and air pollutants emissions and as a result of this study leakage of waste into aquatic environments.

### Estimates of MSW generation, composition, and management by SSP are based on the methodology presented in ref. 2

The country-specific MSW generation per capita is driven by income per capita. Population, urbanization ratio, and income per capita by SSP are taken from SSP Public Database version 2.0. The definition of MSW used in this study is consistent with that in ref. 2 which is the definition adopted in the Directive 851 of the European Parliament and of the Council of 30 May 2018 amending Directive 2008/98/EC on waste[33] (See Supplementary Information S5 for details).

### Estimates of population living up to 1 km

We model the aquatic ecosystems by combining the Hydrorivers dataset[51], the Hydrolakes dataset[52], and the FAO administrative boundaries from which we derive coastlines. Based on these features we use a spatial database to construct buffer zones around rivers, lakes, and coastlines in 250 m intervals up to 1 km in straight line distance. These zones are constructed in a logical way to avoid overlaps with other buffer zones, between countries or water bodies. The population is matched to one of the following in order of priority: a coastline, a lake, or a river. A cell is always attributed to the closer water body of the same type. This ensures that a population cell is only ever attached to a single zone.

Population counts are derived from the JRC GHSL dataset for 2022[53] and classified as either urban or rural based on the Degree of Urbanization[54]. Using spatial overlay, population cells are matched if their midpoint intersects with one of the buffer areas. Urban and rural

**Table 1 | GAINS municipal solid waste matrix**

| Solid waste management technology | Municipal solid waste | | | | | | | |
|---|---|---|---|---|---|---|---|---|
| | Food | Glass | Metal | Other | Paper | Plastic | Textile | Wood |
| Open burned | X | | | X | X | X | X | X |
| Scattered and/or disposed to water-courses | X | X | X | X | X | X | X | X |
| Unmanaged solid waste disposal site - low humidity - <5 m deep | X | | | X | X | | X | X |
| Unmanaged solid waste disposal site - high humidity - > 5 m deep | X | | | X | X | | X | X |
| Compacted landfill | X | X | X | X | X | X | X | X |
| Covered landfill | X | | | X | X | | X | X |
| Landfill gas recovery and flaring | X | | | X | X | | X | X |
| Landfill gas recovery and used | X | | | X | X | | X | X |
| Incineration (poor air quality controls) | X | | | X | X | X | X | X |
| Incineration (high quality air pollution controls - energy recovery) | X | | | X | X | X | X | X |
| Anaerobic digestion | X | | | | | | | |
| Composting | X | | | | | | | |
| Recycling | | X | X | | X | X | X | X |

Source: (ref. 45).

population is attached separately to each of these zones and summarized per unique identifier by region. To speed up processing we exclude smaller lakes with an area of less than 50 km² and rivers with a classical stream order higher than three.

**Estimates of population living up to 1 km up to 2040 in the SSPs**
The share of urban and rural population of the SSPs were aligned to the definition of the JRC to ensure consistency in the calculations. The share of urban and rural population from the JRC in 2020 was applied to the total population by country of the SSPs in 2020. The growth rate of urban population from the SSPs was then applied to the 2020 estimates. Rural population was calculated by deducting urban population from total population.

Combining the new urban and rural population estimates with the spatial dataset of population living up to 1 km from rivers, lakes, and coastal areas in urban and rural areas in 2020, we estimated the number of people living in this buffer zone by year and SSP. We recognize limitations of this approach as we are assuming increases in population density but do not consider area expansion.

**Assessment of potential leakage of MSW into rivers, lakes, and coastal areas**
Estimation of scattered MSW at country/region level for urban and rural areas up to 1 km has been downscaled using the fraction population residing up to 1 km of aquatic ecosystems in urban and rural settings (see point 2 above). We then use the share of population living up to 1 km and apply it to the estimated scattered MSW by type. In that way, scattered MSW gets distributed according to the populated area. We separately estimate the scattered MSW fraction in urban and rural areas by GAINS country/region. Here we assume that a fraction of the mismanaged MSW generated by inhabitants residing close (up to 1 km) to rivers, lakes and coastal areas in urban and rural areas has the potential to reach these aquatic environments. We chose 1 km of MSW fate leakage into aquatic environments based on the three highest levels of fate leakage potential factors in ref. 55, which refer to point sources in close proximity to <1 km to water systems. This approach has been applied to identify the fate, leakage hotspots and management strategies of non-recyclable plastic waste in Indonesia[13]. A distance of 1 km has been used to estimate the vulnerability of mountain rivers to waste dumping in Romania[56] and to estimate rural plastic emissions into the largest mountain lake of the Eastern Carpathians[11]. The assessment of potential leakage of MSW into rivers, lakes and seas

by country/region is estimated applying Eq. (1).

$$LMSW_{(r,l,s)y} = \sum_{i=d,j} \left( MSW_{g(U,R)_y} * UNC_{f(U,R)_y} * SCATT_{f(u,r)_y} * \rho_d * \alpha_j \right) \quad (1)$$

$$d \in \boldsymbol{D}, j \in \boldsymbol{J}$$

Where:

$LMSW_{(r,l,s)y}$ is leakage of MSW in rivers $r$, lakes $l$, seas $s$ in a year $y$
$MSW_{g(U,R)_y}$ is total MSW generation in urban $U$ and rural $R$ areas in a year $y$ estimated as:

$$MSW_{g(U,R)_y} = MSW_{g(U)_y} + MSW_{g(R)_y} \quad (2)$$

$UNC_{f(U,R)_y}$ is the fraction of uncollected waste in urban $U$ and rural $R$ areas in a year $y$ estimated using Eq. (3).

$$UNC_{f(U,R)_y} = \frac{MSW_{g(U)_y} * \left(1 - COLL_{f(U)y}\right) + MSW_{g(R)_y} * \left(1 - COLL_{f(R)y}\right)}{MSW_{g(U)_y} + MSW_{g(R)_y}} \quad (3)$$

$SCATT_{f(u,r)_y}$ is the fraction of uncollected MSW left over an area in urban $U$ and rural $R$ areas in a year $y$ (Eq. (4)). The amount of scattered MSW by country, urban-rural, and type is assessed by subtracting the amount of MSW uncollected openly burned from the total uncollected MSW presented in ref. 2 Supplementary Data[57]. The level of wealth and technological development have a direct impact on the development and improvement of waste management systems[58]. Therefore, an enhancement of waste management systems results in the reduction of scattered and open burning of MSW[59].

$$SCATT_{f(u,r)_y} = MSW_{g(U,R)_y} * UNC_{f(U,R)_y} * (1 - UMSW_{fob(U,R)_y}) \quad (4)$$

$\rho_d$ is the share of population $\rho$ residing in distance $d$ and $\alpha_j$ is the fate of MSW leakage according to the distance from rivers, lakes, and coastal areas up to 1 km.

$$\left.\begin{cases} d_1 = 0m - 249m, & \alpha_1 = 0.85 \\ d_2 = 250m - 449m, & \alpha_2 = 0.75 \\ d_3 = 500m - 749m, & \alpha_3 = 0.55 \\ d_4 = 750m - 1000m, & \alpha_4 = 0.25 \end{cases}\right\}\boldsymbol{D},\boldsymbol{J}$$

## Development of scenarios

The scenarios linked to the five SSPs describe plausible future developments of the MSW management at global level. *Baseline scenarios* represent the MSW administration systems under current legislation (CLE) meaning that not additional policies are adopted until 2040. For each of the *baseline scenarios* a *mitigation scenario* (MFR) is developed according to the SSPs narratives[28]. The *mitigation scenarios* assume that it will be possible to globally adopt circular MSW management systems according to the EU's waste management hierarchy (Directive 2008/98/EC)[6]. The ability to implement circular MSW systems is consistent with the SSPs narratives, in which mitigation and adaptation challenges vary depending on the pathway. Note that the mitigation scenarios explore the technical frontier without considering implementation costs or future economic incentives. Supplementary Information S6 presents a detailed description of the *mitigation scenario* narratives.

## Uncertainty and limitations of the study

Our assessment of MSW generation, composition, and management is based on ref. 2, thereby carries forward the uncertainties there described. Due to the lack of information and inconsistency in the reporting, the uncertainty associated with waste composition and management is not estimated here. Other datasets such as population information, rivers, lakes, and coastal are exogenous to our model and we do not estimate the uncertainties associated with those models. Instead, our results focus on the uncertainty associated with our MSW generation model. Furthermore, estimates concerning the fractions of scattered MSW are highly uncertain as specific information is in many cases not available and assumptions are necessary to cover that gap. We estimate MSW leakage potential into aquatic environments based on the quantities of uncollected waste. We are however aware, that MSW leakage can occur during the collection and transport of MSW as well as in dumpsites. We ran a Montecarlo simulation (1000) and performed a sensitivity analysis (100 samples) using the packages *sensitivity, randtoolbox,* and *lhs* in R studio. In our study we use the coefficient of variation to identify input parameters with higher relative variability. The results indicate that uncollected waste is the variable with a potentially more significant impact on waste leakage (Supplementary Information S9 shows the average CV for the simulations). In addition, note that we apply the same fate factors for all MSW streams, however, we are aware that different streams may behave different according to the size, volume, density, degradability and may also be influenced by climatic and geographical conditions in a different way. Furthermore, we consider the share of urban and rural population living in a buffer zone of 1 km from rivers, lakes, and coastal areas based on the GHSL-Global Human Settlement Layer depicting the distribution of population in urban and rural areas for 2020. We applied the same share to all years and scenarios due to the unavailability of gridded population distribution for the different SSPS and years in study. This may lead to under or over estimation of MSW leakage in urban and rural areas depending on the scenario. Other limitations such as the accuracy of the used spatial datasets, mainly the JRC GHSL and Hydrosheds datasets, apply as per their respective description. The exclusion of smaller lakes should have little effect at most as these water bodies are still reflected in the Hydrorivers dataset and their extent will partly be reflected by the buffer zones.

While the databases and information we use represent the best available to our knowledge on a global level, we are aware that this analysis cannot fully capture the complexity of real-world scenarios. The data and methods applied each come with their own limitations, such as data quality and availability or characteristics of the physical world which could not be modeled. Aspects such as future variations in environmental conditions, geopolitical unrest or unforeseen events could limit the applicability of the results additionally. This underlines the importance of developing and maintaining Monitoring, Reporting, and Verification frameworks to validate and adapt the analysis, aligning it further with the development of the world. Irrespective of the uncertainties and limitations, we demonstrate that to curtail leakage of MSW into aquatic environments will only be possible if global circular MSW management systems are adopted.

## Reporting summary

Further information on research design is available in the Nature Portfolio Reporting Summary linked to this article.

## Data availability

All data generated during this study is included in this published article (and its Supplementary Information). The Supplementary Data 1 generated in this study has been deposited in: https://doi.org/10.6084/m9.figshare.23855370. Source data are provided with this paper. Databases used SSP Public Database version 2.0. Population, GDP and Urbanization data. https://tntcat.iiasa.ac.at/SspDb/dsd?Action=htmlpage&page=about, GHSL-Global Human Settlement Layer Population Count: https://doi.org/10.2905/D6D86A90-4351-4508-99C1-CB074B022C4A GHSL-Global Human Settlement Layer Degree of Urbanization: https://doi.org/10.2905/4606D58A-DC08-463C-86A9-D49EF461C47F FAO (Administrative Boundaries and Coastlines): https://data.apps.fao.org/map/catalog/srv/eng/catalog.search#/metadata/9c35ba10-5649-41c8-bdfc-eb78e9e65654 http://cidportal.jrc.ec.europa.eu/ftp/jrc-opendata/GHSL/GHS_STAT_UCDB2015MT_GLOBE_R2019A/V1-2/ GRDC (2020): WMO Basins and Sub-Basins/Global Runoff Data Centre, GRDC. 3rd, rev. ext. ed. Koblenz, Germany: Federal Institute of Hydrology (BfG). https://www.bafg.de/GRDC/EN/02_srvcs/22_gslrs/223_WMO/wmo_regions_node.html HydroLakes: https://www.hydrosheds.org/products/hydrolakes /https://doi.org/10.1002/hyp.9740 HydroRivers: https://www.hydrosheds.org/products/hydrorivers PostGIS,PostGIS; http://postgis.net/ PostreSQL,PostgreSQL; http://www.postgresql.org/ Source data are provided with this paper.

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

## Acknowledgements

The development of this research was supported by the Innovation and Bridging Grants Fund (IBGF) from the International Institute for Applied Systems Analysis A.G.S.

## Author contributions

A.G.S. designed the study, developed the scenarios, performed the data analysis, and prepared the manuscript. F.L. performed the geospatial calculations. A.G.S. and F.L. contributed to writing and revising the manuscript.

## Competing interests

The authors declare no competing interests.
