## [Peer Review File · Nature Communications]

The crucial role of circular waste management systems in cutting waste leakage into aquatic environmentsReviewers' Comments:

Reviewer #1:

Remarks to the Author:

Overall, a good paper. The topic is interesting and debated by the political and scientific community in our days and it concerns to the global issue of waste leakage into aquatic environments. The authors' distinctive approach is the combination of spatial analysis with the shared socioeconomic pathways storylines to develop plausible future waste leakage mitigation strategies up to 2040, and that result in a good graphic visualization of the main waste leakage "hotspots" worldwide, which could be an important basis for strategic and political decision-making, at international and regional levels, of priority actions designed to mitigate waste leakage problem. The results show that although the leakage of waste to the aquatic environments is highly dependent on variables such as population size, length and area of aquatic systems, waste generation, composition and management systems, it is not necessarily the countries with the highest amounts of mismanaged waste that have the highest amounts of waste leakage in the aquatic environments. Another conclusion is that even in a scenario of adopting the best waste management strategies to reduce the entry of waste into aquatic ecosystems, it would be impossible to eliminate waste leakage before 2030, that is, to achieve the waste-related Sustainable Development Goals.

The manuscript is well written but I don't quite agree with the way it is organized, either by the sequence of the main sections or by the fact that some are numbered and others are not. Although this is the indication given to the authors in the guide for submission to Nature Communications, the sequential organization in Introduction followed by sections Results, Discussion and finally Methods, is a little incomprehensible, since the reader needs to previously know the methods used, namely the sources of information or databases used, the defined criteria or assumptions, calculation formulas, etc. to better understand the results and conclusions. If it is possible to change this order this was a recommendation and a request that I make to the authors.

In general terms, the abstract represents a good summary of the research. The introduction appears to be complete and referred to scientific literature, but other papers, projects or organizations that have developed methodologies to quantify waste leakage into the environment could have been referenced, for example the IUCN projects for the quantification and reduction of plastic leakage. The methodology adopted by the authors is reliable, although with the limitations pointed out by the authors. The results and discussion are well explained and well supported by figures, and the authors also provide a lot of supplementary information and very useful data for the overall understanding the results. The conclusions are supported by the results, but the authors could elaborate a little more on the implications that their results may have for decision-makers responsible for international and regional strategies and policies to reduce waste leakage, namely for the global problem of plastic leakage for aquatic ecosystems.

Some aspects that are suggested to be reviewed:

- In the introduction, values or percentages are indicated for "waste generation is expected" (line 60), or "global municipal solid waste generated is mismanaged" (line 64), but it is not indicated if these values refer to all countries, OECD countries, etc. please give this indication; same comment for line 113 (results section);
- Yet in the Introduction, although the term "Marine debris" can also be used, "marine litter" is more common, as it is more associated with littering behaviors (e.g. lines 71, 77);
- In the results leave a space between the number and the unit symbol, e.g. 1km not 1km (the same lines 227 and 352)
- In the paragraph that goes from lines 116 to 1129, for Europe, in addition to the EU New Circular Economy Action Plan, and the directives that are indicated, it would also be important to mention the European Strategy for Plastics in a Circular Economy (2018), the Directive (EU) 2019/904 (also known as the Single Use Plastics Directive) and the initiatives Circular Plastics Alliance and the European Plastic Pact;
- On the line 221 it says "80.8 Mt of MSW leaked into aquatic environments" but if I understand correctly, the same treatment is given to all MSW components, without taking into account, for example, that the behavior of the food is different from the lighter components, such as plastic that is

easily transported by the wind over distances often greater than 1 km. Could you please explain better how you approached this issue? If this has not been taken into account, this limitation should be included in the section "Uncertainty and limitations of the study".

In conclusion, in my opinion the article is of interest and quality for the reason that I recommend it for publication, with the possibility of reorganizing the sessions and with the small amendments indicated.

Reviewer #2:

Remarks to the Author:

The paper has an interesting approach but the discussion/results related to literature review need to be further developed plus additional data to be made available (e.g, scattered waste of MSW per country)

Did you use scattered waste for major geographical area not country level according to data presented in the SI of this manuscript. This should be pointed out as a research limitation because it NOT underpins the regional disparities in terms of MSW performances inside the EU for example. A country could have full waste collection coverage 100 %- but not all waste is collected how scattered waste solve this issue? There are such information at country level in the SI? The Lebreton and Andrady (<https://www.nature.com/articles/s41599-018-0212-7>) calculate the mismanagement plastic waste at country level however these needs to be adjusted to current waste management realities. Also, waste generation data based on GDP are relevant at regional scales? What are limitations with this approach without using waste statistics about MSW/plastic waste flows (generation/collection from national environmental statistics)

Without an improvement of national/regional reliable waste statistics data how can we evaluate the progress towards CE ---these aspects should be better pointed out in the manuscript text

The literature review and discussion should be improved and updated.

How you findings relate with other global studies :

<https://www.science.org/doi/10.1126/science.aba9475>

<https://www.science.org/doi/10.1126/science.aba3656> ,

<https://www.science.org/doi/10.1126/sciadv.abd0288>

Regional context for Central and Eastern Europe about freshwater plastic pollution (Carpathian Ecoregion)

<https://www.sciencedirect.com/science/article/abs/pii/S0048969723026797?via=ihub>

pag 5 Line 115-116: also home composting divert some food waste (vegetables, fruits) from leaking into natural environment

Line 144-145: The composition of scattered waste is estimated to be mostly food (52%), followed by mixed waste (12%), plastic (10%), paper (8%), glass (5%), metal (3%) and other materials (10%). based on what – please give further details, relevant regional differences to be mentioned ?

Page 5: Scattered MSW, defined here as the fraction of MSW not collected and left over an area –

OK. Where such data is available at country level, inside the buffer of 1km etc ? How these data are aggregated/calculated at wider geographical areas presented in figures (EU-28, MEAST)

The amount of scattered MSW by country - which is key parameter in your analysis – should be also available in xls format ? like SI of : www.nature.com/articles/s41599-018-0212-7 in case of MPW for each country. The ref 1 contains some info in SI materials about emissions , there is no data for each country but aggregated ones to wider geographical regions (eg. EU-28).

Otherwise, how someone could check the data in the eq.3 for example ?

Narrative scenarios - the first scenario provides some concrete clues about the plastic waste management (e.g maximum municipal plastic waste rate reduction of 50% by the year 2030 as a part of the 2030 Sustainable Development Goals.) but other lack any key values related to plastics to help

the reader to understand your approach

Therefore, this manuscript need to provide further details and clarifications plus better link with other studies.

Reviewer #3:

Remarks to the Author:

1. This paper tries to evaluate a global water pollution issue driven by illicit or inappropriate waste dumping from receiving water body perspective. The authors claim novelty by using an existing concept and method of "shared socioeconomic pathways".

2. The authors claimed that "To our knowledge, no global assessment exists that combines the Shared Socio-economic Pathways (SSPs) with waste generation and management storylines and spatial analysis to analyze how the adoption of circular waste management systems in urban and rural areas can cut leakage of waste into aquatic environments (lakes, rivers and coastal areas)." However, such concept can be seen in "O'Neill, B. C. et al. The roads ahead: Narratives for shared socioeconomic pathways describing world futures in the 21st century. *Global Environmental Change* 42, 169–180 (2017)." Simply applied an existing approach to deal with a particular sector is not significant enough for a decent journal publication.

3. The authors claimed that "We distinguish between rural and urban areas under five future socioeconomic pathways up to 2040. The results of this study can be further developed in combination with a hydrological transport model to estimate the amounts of MSW potentially reaching the oceans and their estimated origin. But there is no detail about the types of hydrological transport models they chose and how can they calibrate the models around 150+ countries in the world. I think a lot of discussion herein is based on speculation and impression rather than rigorous reasoning with quantitative evidence.

4. It is unclear that on what basis the Fig. 4 can have a reliable presentation for overview of the ten highest potential MSW sources of each type, sized relative within their type.

5. Mitigation scenarios lack managerial insight, economic incentives, and/or policy contexts. It looks like a sparse discussion with no substance in MSW management. It sounds like a chatting room discussion.

6. Overall, the paper was poorly structured, streamlined, and written. There are too many grammar errors, wording problems, poor sentence structures, and unreasonable logic flows in English writing. The readability of this paper is very low. With is said, I cannot recommend the paper be published by Nature-Communication.

Comments to the author	Response from the author
Reviewer #1	
Overall, a good paper. The topic is interesting and debated by the political and scientific community in our days and it concerns to the global issue of waste leakage into aquatic environments. The authors' distinctive approach is the combination of spatial analysis with the shared socioeconomic pathways storylines to develop plausible future waste leakage mitigation strategies up to 2040, and that result in a good graphic visualization of the main waste leakage "hotspots" worldwide, which could be an important basis for strategic and political decision-making, at international and regional levels, of priority actions designed to mitigate waste leakage problem. The results show that although the leakage of waste to the aquatic environments is highly dependent on variables such as population size, length and area of aquatic systems, waste generation, composition, and management systems, it is not necessarily the countries with the highest amounts of mismanaged waste that have the highest amounts of waste leakage in the aquatic environments. Another conclusion is that even in a scenario of adopting the best waste management strategies to reduce the entry of waste into aquatic ecosystems, it would be impossible to eliminate waste leakage before 2030, that is, to achieve the waste-related Sustainable Development Goals.	Thank you for this comment. We aimed to generate added value for research.
The manuscript is well written but I don't quite agree with the way it is organized, either by the sequence of the main sections or by the fact that some are numbered and others are not. Although this is the indication given to the authors in the guide for submission to Nature Communications, the sequential organization in Introduction followed by sections Results, Discussion and finally Methods, is a little	Thank you for this suggestion. We have revised the manuscript, thereby aiming to reduce jargon and make the text more readable throughout. After consulting with the editor, it is unfortunately not possible to change the section order. However, we have added a short preamble in the different sections outlining the methods which hopefully addresses the issue.

incomprehensible, since the reader needs to previously know the methods used, namely the sources of information or databases used, the defined criteria or assumptions, calculation formulas, etc. to better understand the results and conclusions. If it is possible to change this order this was a recommendation and a request that I make to the authors.	
In general terms, the abstract represents a good summary of the research. The introduction appears to be complete and referred to scientific literature, but other papers, projects or organizations that have developed methodologies to quantify waste leakage into the environment could have been referenced, for example the IUCN projects for the quantification and reduction of plastic leakage	Thanks for suggesting adding the work carried out from different institutions. This adds relevant information about actions happening on the ground towards reducing plastic waste leakage. We have added the following sentence in the introduction of the manuscript: “Initiatives to reduce plastic waste leakages include the work and economic impact analyses of plastic pollution carried out by the International Union for Conservation of Nature (IUCN) in Fiji¹³, Samoa¹⁴, Vanuatu¹⁵, Antigua and Barbuda¹⁶, the Mediterranean islands ¹⁷, among others”. Furthermore, we have also included a short paragraph mentioning the different scopes and methodologies to assess plastic waste leakage. The paragraph reads: “It is important to note that the scope and methodologies to estimate plastic waste leakage differ. While some methodologies include macroplastics from production to use and fate ²⁰ or are based on population and spatial analysis²¹, others assess micro – and macro plastic waste leakage over the entire life cycle of a product (corporate plastic footprint)²²”.
The methodology adopted by the authors is reliable, although with the limitations pointed out by the authors. The results and discussion are well explained and well supported by figures, and the authors also provide a lot of supplementary information and very useful data for the overall understanding the results. The conclusions are supported by the results, but the authors could elaborate a little more on the implications that their results may have for decision-makers responsible for international and regional strategies and policies to reduce waste leakage, namely for the global problem of plastic leakage for aquatic ecosystems.	Thanks for your comment. We have worked on linking the results and conclusions to the strategies and actions that can be taken at political level to reduce MSW leakage.

Some aspects that are suggested to be reviewed:  - In the introduction, values or percentages are indicated for "waste generation is expected" (line 60), or "global municipal solid waste generated is mismanaged" (line 64), but it is not indicated if these values refer to all countries, OECD countries, etc. please give this indication; same comment for line 113 (results section); 	Thanks for noticing this. Indeed, the specification was missing. We have added “global municipal waste generation” . We have also specified that the assessment refers to global municipal solid waste” We have added the following clarification in the introduction and in the methods: “The The IIASA-GAINS model is used as a framework to carry out this assessment. The GAINS model has global coverage with a geographic representation of 180 country/regions with multitemporal resolution at five years intervals. The MSW sector in the model further differentiates between urban and rural areas within a country/region”.
 - Yet in the Introduction, although the term "Marine debris" can also be used, "marine litter" is more common, as it is more associated with littering behaviors (e.g., lines 71, 77); 	Thanks for pointing out this. We have revised the definition of marine debris and decided to accept your suggestion. We use now ‘marine litter’ throughout the text. The Definition of marine debris is: “Litter disposal and accumulation in the marine environment is one of the fastest-growing threats to the health of the world's oceans (Pham et al., 2014). Marine debris, also known as marine litter, has been defined by UNEP (2009) as “any persistent, manufactured or processed solid material discarded, disposed of or abandoned in the marine and coastal environment”. Marine debris consists of items that have been made or used by people and deliberately discarded into the sea or rivers or on beaches; brought indirectly to the sea with rivers, sewage, storm water or winds; accidentally lost, including material lost at sea in bad weather (fishing gear, cargo); or deliberately left by people on beaches and shores (UNEP, 2005)”.
 - In the results leave a space between the number and the unit symbol, e.g., 1km not 1km (the same lines 227 and 352) 	Thanks for noticing this. We now leave a space between the number and the unit.
 -- In the paragraph that goes from lines 116 to 1129, for Europe, in addition to the EU New Circular Economy Action Plan, and the directives that are indicated, it would also be important to mention the European Strategy for Plastics in a Circular Economy (2018), the Directive (EU) 2019/904 (also known as the Single Use Plastics Directive) and the initiatives Circular Plastics Alliance and the European Plastic Pact; 	Thanks for suggesting this. We have amended the paragraph as follows: “MSW leakage in rivers (and in all other aquatic ecosystems) already in the Baseline Scenario resulting from the implementation of policies and strategies tackling improvements in waste management rather than in reduction of waste generation. These policies include the EU Waste Framework Directive 2008/98/EC³² and the amendment EU Directive 2018/851³³, the EU Directive on packaging and packaging waste 94/62/EC³⁴ and the amendment EU Directive 2018/852³⁵, the EU Directive 2019/904³⁶, the EU New Circular Economy Action Plan (which includes the European Strategy for Plastics)³⁷, the Circular Plastic Alliance³⁸, the European Plastic Pact³⁹, the 3R’s strategy in Japan⁴⁰ and the National Waste Policy Action Plan 2019 in Australia⁴¹”.

	For clarification: https://environment.ec.europa.eu/strategy/plastics-strategy_en “...The EU adopted a European strategy for plastics in January 2018. It is part of the EU’s circular economy action plan, and builds on existing measures to reduce plastic waste...”
- On the line 221 it says "80.8 Mt of MSW leaked into aquatic environments" but if I understand correctly, the same treatment is given to all MSW components, without taking into account, for example, that the behavior of the food is different from the lighter components, such as plastic that is easily transported by the wind over distances often greater than 1 km. Could you please explain better how you approached this issue? If this has not been taken into account, this limitation should be included in the section "Uncertainty and limitations of the study".	Thanks for highlighting this. We have added the following clarification at the beginning of the section Results: Global identification of potential MSW leakage in aquatic ecosystems and mitigation scenarios: “Identification of potential MSW reaching rivers, lakes and coastal areas is based on the amount of uncollected scattered MSW by waste stream generated by the population residing at a distance up to 1 km from the aquatic systems. We divide the distance (0 – 1 km) in four buffer zones of 250 m each. We then apply different factors for each buffer zone representing the fate of MSW leakage according to the distance from rivers, lakes, and coastal areas distinguishing between urban and rural areas (see Methods). Note that we apply the same fate factors for all streams, however, we are aware that different streams may behave different according to the size, volume, density, degradability and may also be influenced by climatic and geographical conditions in a different way”.
Reviewer #2	
The paper has an interesting approach, but the discussion/results related to literature review need to be further developed plus additional data to be made available (e.g, scattered waste of MSW per country)	Thank you for the encouraging feedback. We have worked on improving the discussion related to the literature. We have provided information at GAINS country/region level in the Supplementary Dataset. The information includes total MSW generated, scattered MSW, MSW reaching rivers, lakes, and coastal areas, disaggregated by fraction (e.g., food, paper, plastic, etc.) and urban- rural areas.
Did you use scattered waste for major geographical area not country level according to data presented in the SI of this manuscript. This should be pointed out as a research limitation because it NOT underpins the regional disparities in terms of MSW performances inside the EU for example.	Thanks for requiring a clarification on this. We have added the following clarification in the introduction and in the methods: “The IIASA-GAINS model is used as a framework to carry out this assessment. The GAINS model has global coverage with a geographic representation of 180

	country/regions with multitemporal resolution at five years intervals. The MSW sector in the model further differentiates between urban and rural areas within a country/region". Table S4 in the Supplementary Information displays the GAINS country/region included in the analysis and how we aggregated them by region". We have provided information at GAINS country/region level in the Supplementary Dataset. The information includes total MSW generated, scattered MSW, MSW reaching rivers, lakes, and coastal areas, disaggregated by fraction (e.g., food, paper, plastic, etc) and urban- rural areas.
A country could have full waste collection coverage 100 %- but not all waste is collected how scattered waste solve this issue? There are such information at country level in the SI? The Lebreton and Andrady (https://www.nature.com/articles/s41599-018-0212-7) calculate the mismanagement plastic waste at country level however these needs to be adjusted to current waste management realities.	Thanks for this comment and providing this important reference. Indeed, there are countries (e.g., Switzerland, Luxemburg, Austria, among others) that having 100% collection coverage still have waste leakage. Recommendations to reduce this fraction of waste leakage have been published in the report: From source to sea – The untold story of marine litter from the European Environment Agency (https://www.eea.europa.eu/publications/european-marine-litter-assessment). Most of recommendations are related to improvements of the monitoring of litter by combining diverse data sources and improve riverine monitoring. It also includes modelling tools (for example this work) to better predict and identify hotspots and areas of action. We have referenced Lebreton’s et al. work, including the articles: River plastic emissions to the world’s oceans. Nature Communications 8, 15611 (2017) (Ref 18 in the manuscript) and Meijer, L. J. J., van Emmerik, T., van der Ent, R., Schmidt, C. & Lebreton, L. More than 1000 rivers account for 80% of global riverine plastic emissions into the ocean. Science Advances 7, eaaz5803 (2019). We have added this additional reference to complement the work that has been done by these authors in that respect in the main manuscript as well as in the supplementary material Table S1. Studies assessing leakage of waste into aquatic environments. We have included this information in the Supplementary Dataset at the GAINS country/region level.

Without an improvement of national/regional reliable waste statistics data how can we evaluate the progress towards CE --these aspects should be better pointed out in the manuscript text	Thanks for pointing out this relevant aspect. We have added a statement in the conclusion referring to the improvement of MSW reporting frameworks as follows: “Therefore, actions need to be taken to develop a standardized reporting framework that can support the monitoring of MSW generation, composition, and flows, and follow up the implementation of actions (including political, economic, and technological measures) targeted to the reduction of MSW and improvement of waste management systems. A standardized framework will reduce the uncertainty of the assessments and will provide better knowledge and information to develop strategies and take actions at different levels (i.e., regional, urban-rural, streams) to tackle the MSW crises. This framework can also contribute to monitor the progress of the circular economy regarding availability and flows of secondary materials.”
The literature review and discussion should be improved and updated. How you findings relate with other global studies : https://www.science.org/doi/10.1126/science.aba9475 https://www.science.org/doi/10.1126/science.aba3656 , https://www.science.org/doi/10.1126/sciadv.abd0288 Regional context for Central and Eastern Europe about freshwater plastic pollution (Carpathian Ecoregion) https://www.sciencedirect.com/science/article/abs/pii/S0048969723026797?via=ihub	Thanks a lot for providing these valuable references. They have helped us to update the literature and have provided interesting insights for the discussion. We have included them in the manuscript and supplement. The article related to the Carpathian region actually complements an earlier article about rural plastic waste leakage in the Easter Carpathians. The paragraph now reads: “Regional studies include an analysis of rural plastic emissions into the Izvoru Muntelui lake (Eastern Carpathians) which suggests that rural municipalities might be responsible for 85.51% of plastic bottles collected between 2005 and 2010 and it concludes that plastic pollution is mainly local¹². A more recent study on the Carpathian region identifies that watercourses below 750 m.a.s.l are significantly affected by mismanaged plastic waste and most of the hotspots are located in Romania, Hungary, and the Ukraine¹³”. (Liro, M. et al. Mountains of plastic: Mismanaged plastic waste along the Carpathian watercourses. Science of The Total Environment 888, 164058 (2023)). “A recent assessment demonstrates that United States generated the largest amount of plastic waste in 2016, of which between 0.14 and 0.41 Mt was illegally dumped and 0.15 to 0.99 Mt was exported as recycling material that ended up being inappropriate

	managed¹⁴ (Law, K. L. et al. The United States' contribution of plastic waste to land and ocean. Science Advances 6, eabd0288). The suggested references are included in the summary Table S1. Studies assessing leakage of waste into aquatic environments.
pag 5 Line 115-116: also home composting divert some food waste (vegetables, fruits) from leaking into natural environment	Thanks for sharing this comment. We have added this important aspect in the text. Now it reads: “This estimate assumes that part of the uncollected food waste in rural areas in developing countries is used as animal feed²⁴ or composted at household level²⁵ and therefore is not accounted for as scattered waste.”
Line 144-145: The composition of scattered waste is estimated to be mostly food (52%), followed by mixed waste (12%), plastic (10%), paper (8%), glass (5%), metal (3%) and other materials (10%). based on what – please give further details, relevant regional differences to be mentioned ?	Thanks for requiring further details on this. We have added a figure displaying the composition of scattered waste in the supplement Fig S2 and the following paragraphs in the manuscript: “MSW generation and composition is estimated by using different elasticities representing four different income averages assuming that MSW composition is dependent on average national income levels as stated in Gómez-Sanabria et al., 2022¹. Global MSW generation is estimated at about 2560 Mt (million ton) in 2020¹ and it is expected to increase to 3320–3790 Mt in 2040 depending on the followed socio-economic pathway¹. The estimates show that in 2020 the world generated 1091 Mt of food waste (43% of MSW), 260 Mt of plastic waste (10%), 366 Mt of paper waste (14%), 113 Mt of glass waste (4%), 73 (3%) Mt of metal waste 651 Mt (26%) of other waste (including, textile, wood, and mixed waste). The results by stream are in line with those assessed by Chen et al (2020)²⁵ and with Lebreton et al., (2019) for plastic waste (239 Mt)²¹ in 2020. Future average global composition of MSW will see a slight decline of organic waste fraction (food) in all SSPs, except in the SSP3. Nonetheless, food waste will remain as the highest portion of MSW in the future across all SSPs in absolute terms, finding that is in agreement with Chen et al., (2020)²⁵. In 2040, food waste is expected to increase between 26% and 40%, plastic waste between 37% and 45% and paper waste between 27% and 50% depending on the socio-economic pathway when compared to 2020 quantities. When looking at the composition of MSW the SSP1_MFR in 2040 stands out with the share of food waste being reduced to 28% due to implementation of the food waste reduction target. Future paper waste fraction may increase due to the reduction of plastic under the assumption that the policy reducing plastic waste is adopted. Our results show that other waste streams can increase as a potential rebound effect of this measure (e.g., paper cups replacing single-use plastic cups). Therefore, targets to reduce other waste streams are also urgently needed.

	The amount of scattered MSW, defined here as the fraction of MSW not collected and left over an area, is assessed for each stream (i.e., food, plastic, paper, etc.) by subtracting the amount of MSW open burned from the uncollected MSW (see Methods). Our results suggest that scattered MSW accounts for about 14% or 350 Mt of the total global MSW generation in 2020 (Fig. 1). This estimate assumes that part of the uncollected food waste in rural areas in developing countries is used as animal feed²⁶ or composted at household level²⁷ and therefore is not accounted for as scattered waste. Over 87% of the total scattered MSW is generated across China (30%), South Asia (20%), Africa (20%) and India (17%). Wealthier countries are associated with lower levels of mismanaged MSW² and hence their contribution to total scattered MSW is comparatively lower. The composition of scattered MSW in 2020 is assumed to be 52% (181 Mt) food, 9% (33 Mt) plastic, 8% (27 Mt) paper, 18 Mt (5%) glass, 11 Mt (3%) metal and 78 Mt (23%) other waste. The resulting composition of scattered waste is therefore highly depending on the levels of collection rates as well as type of management by stream. In regions such as Africa, South Asia, China, and Latin America and the Caribbean (LCAM), food waste accounts for the highest fraction of scattered waste (between 49% and 53% of the total MSW) while in regions like EU27 + UK and Russia, plastic and glass waste make up the majority of scattered waste. The proportion of paper waste in North America and Oceania OECD and mixed waste in India in the scattered waste fraction is substantial (see supplement Fig S1 and Fig S2)”.
Page 5: Scattered MSW, defined here as the fraction of MSW not collected and left over an area – OK. Where such data is available at country level, inside the buffer of 1 km etc ? How these data are aggregated/calculated at wider geographical areas presented in figures (EU-28, MEAST)	Thank you for your helpful comment. We have added a paragraph in the results and methods section for clarification: Identification of potential MSW reaching rivers, lakes and coastal areas is based on the amount of uncollected scattered MSW by waste stream generated by the population residing at a distance up to 1 km from the aquatic systems. We divide the distance (0 – 1 km) in four buffer zones of 250 m each. We then apply different factors for each buffer zone representing the fate of MSW leakage according to the distance from rivers, lakes, and coastal areas distinguishing between urban and rural areas (see Methods). Note that we apply the same fate factors for all streams, however, we are aware that different streams may behave different according to the size, volume, density, degradability and may also be influenced by climatic and geographical conditions in a different way.

	4. Assessment of potential leakage of MSW into rivers, lakes and coastal areas Estimation of scattered MSW at country/region level for urban and rural areas up to 1 km has been downscaled using the fraction population residing up to 1 km of aquatic ecosystems in urban and rural settings (see point 2 above). We then use the share of population living up to 1 km and apply it to the estimated scattered MSW by type. In that way, scattered MSW gets distributed according to the populated area. We separately estimate the scattered MSW fraction in urban and rural areas by GAINS country/region. Table S4 in the supplementary information presents the GAINS country/regions included in this assessment and maps the GAINS countries/regions to the regional aggregation.
The amount of scattered MSW by country - which is key parameter in your analysis – should be also available in xls format ? like SI of : www.nature.com/articles/s41599-018-0212-7 in case of MPW for each country. The ref 1 contains some info in SI materials about emissions , there is no data for each country but aggregated ones to wider geographical regions (eg. EU-28). Otherwise, how someone could check the data in the eq.3 for example ?	Thanks for requesting additional data. We have provided information at GAINS country/region level in the Supplementary Dataset. The information includes total MSW generated, scattered MSW, MSW reaching rivers, lakes, and coastal areas, disaggregated by fraction (e.g., food, paper, plastic, etc) and urban- rural areas.
Narrative scenarios - the first scenario provides some concrete clues about the plastic waste management (e.g maximum municipal plastic waste rate reduction of 50% by the year 2030 as a part of the 2030 Sustainable Development Goals.) but other lack any key values related to plastics to help the reader to understand your approach Therefore, this manuscript need to provide further details and clarifications plus better link with other studies.	Thanks for pointing this out. We have included in the supplement (S6) tables describing each SSP in terms of socio-economic development, environmental challenges, and improvements in waste management systems. In addition, we added the following paragraph in the main text: Our mitigation scenarios adopt the socio-economic narratives from the Shared Socioeconomic Pathways (SSPs) (see O’Neill et al, 2017²⁷). The SSPs provide five plausible pathways about probable world’s socioeconomic development. Based on our interpretation of the narratives, we develop circular MSW management scenarios representing mitigation and/or adaptation challenges. The implementation of circular waste management systems are develop in accordance with the EU’s waste management hierarchy – Directive 2008/98/EC³¹ as stated in Gómez-Sanabria et al., (2022)¹. A circular waste management system is defined here as a system with successful implementation of MSW reduction policies by reducing food and plastic waste generation, maximum

	technical feasible recycling rates of all MSW streams, and once recycling capacity is exhausted, incineration of refuse MSW with energy recovery. Furthermore anaerobic digestion is implemented to treat food and garden waste, high diversion of MSW from landfills and upgrading of dumpsites⁴³. A description of the narratives in terms of economic development and demographics for each SSPs along with the description of MSW management scenarios is presented in the supplement S6.
Reviewer #3	
1. This paper tries to evaluate a global water pollution issue driven by illicit or inappropriate waste dumping from receiving water body perspective. The authors claim novelty by using an existing concept and method of "shared socioeconomic pathways".	Thank you for the comment.
2. The authors claimed that “To our knowledge, no global assessment exists that combines the Shared Socio-economic Pathways (SSPs) with waste generation and management storylines and spatial analysis to analyze how the adoption of circular waste management systems in urban and rural areas can cut leakage of waste into aquatic environments (lakes, rivers and coastal areas).” However, such concept can be seen in “O’Neill, B. C. et al. The roads ahead: Narratives for shared socioeconomic pathways describing world futures in the 21st century. Global Environmental Change 42, 169–180 (2017).” Simply applied an existing approach to deal with a particular sector is not significant enough for a decent journal publication. Everyone can make a new storyline by implementing an existing method to a particular problem in the sense that she or he will have 1 million more publications in the future. This is not research. Instead, it is just a project in practice.	Thanks for your comment. We understand your concern about misusing the SSPs for this kind of assessments. However, we believe that socio-economic scenarios are an important tool to assess the impacts of human activity on the environment. The SSPs allow researchers to explore alternative responses to mitigate and adapt to future alternative world developments. For example, the SSPs are an important input for the recent and ongoing IPCC Assessment Reports and are central to the climate research community. Furthermore, different sectors (energy, transport, agriculture, waste, etc.) will be affected in diverse ways depending on the pathway. Actions to adapt or mitigate the impact will happen at different levels and scales depending on the sector. Please find below some relevant studies adopting the SSPs as a framework for different sectors: Marina Andrijevic et al 2021. Future cooling gap in shared socio-economic pathways. Environmental Research Letters.16 094053 https://doi.org/10.1088/1748-9326/ac2195 Kikstra e al., 2021. Decent living gaps and energy needs around the world. Environmental Research Letters. 16. 09. https://topscience.iop.org/article/10.1088/1748-9326/ac1c27

	Strokal et al., 2021. Urbanization: an increasing source of multiple pollutants to rivers in the 21st century. Urban Sustainability. 24. https://www.nature.com/articles/s42949-021-00026-w Mitter et al., 2020. Shared Socio-economic Pathways for European agriculture and food systems: The Eur-Agri-SSPs. Global Environmental Change. 102159. 65. https://doi.org/10.1016/j.gloenvcha.2020.102159
The authors claimed that “We distinguish between rural and urban areas under five future socioeconomic pathways up to 2040. The results of this study can be further developed in combination with a hydrological transport model to estimate the amounts of MSW potentially reaching the oceans and their estimated origin. But there is no detail about the types of hydrological transport models they chose and how can they calibrate the models around 150+ countries in the world. I think a lot of discussion herein is based on speculation and impression rather than rigorous reasoning with quantitative evidence.	Thanks for your comment. We do not estimate the amounts of MSW potentially reaching (transported to) oceans as this is out of the scope of our study. Instead, we attempt to identify the “hotspots” where MSW is at high risk of entering water courses. To avoid confusions, we have modified the sentence as follow: “The results of this study can be further developed in combination with additional environmental, meteorological and geographical variables, as demonstrated in Meijer et al., (2019)⁴ who included characteristics such as slope, precipitation, stream order, and river discharge to estimate the amounts of MSW potentially reaching the oceans as well as their estimated origin”.
4. It is unclear that on what basis the Fig. 4 can have a reliable presentation for overview of the ten highest potential MSW sources of each type, sized relative within their type.	Thank you for your feedback. We have added a supplementary table describing the basis of Figure 4 including ancillary information. This is meant as a broad overview to show hotspots for the three different types of aquatic systems assessed by our study and regional differences which are discussed in the section “Identification of potential MSW leakage in aquatic ecosystems”.
5. Mitigation scenarios lack managerial insight, economic incentives, and/or policy contexts. It looks like a sparse discussion with no substance in MSW management. It sounds like a chatting room discussion.	Thanks for pointing this out. We have included in the supplement (S6) tables describing each SSP in terms of socio-economic development, environmental challenges, and improvements in waste management systems. In addition, we added the following paragraph in the main text: Our mitigation scenarios adopt the socio-economic narratives from the Shared Socioeconomic Pathways (SSPs) (see O’Neill et al, 2017²⁸). The SSPs provide five

	plausible pathways about the world's probable socioeconomic development. Based on our interpretation of the narratives, we develop circular MSW management scenarios representing mitigation and/or adaptation challenges. The implementation of circular waste management systems are developed in accordance with the EU's waste management hierarchy – Directive 2008/98/EC³² as stated in Gómez-Sanabria et al., (2022)¹. A circular waste management system is defined here as a system with successful implementation of MSW reduction policies by reducing food and plastic waste generation, maximum technically feasible recycling rates of all MSW streams, and once recycling capacity is exhausted, incineration of refuse MSW with energy recovery. Furthermore anaerobic digestion is implemented to treat food and garden waste, high diversion of MSW from landfills and upgrading of dumpsites⁴⁵. A description of the narratives in terms of economic development and demographics for each SSPs along with the description of MSW management scenarios is presented in the supplement S6. Note that mitigation scenarios are represented by an additional MFR in the naming (e.g., SSP1_MFR).
6. Overall, the paper was poorly structured, streamlined, and written. There are too many grammar errors, wording problems, poor sentence structures, and unreasonable logic flows in English writing. The readability of this paper is very low. With is said, I cannot recommend the paper be published by Nature-Communication.	Thank you for your feedback. We have revised the manuscript, thereby aiming to reduce jargon and make the text more readable throughout.

Reviewers' Comments:

Reviewer #2:

Remarks to the Author:

The revised manuscript is better structured with further explanations of results/ discussions related to other studies compared to the initial submission. The additional data and info provided in the SI materials were required to better understand the modeling process, results, and its shortcomings. The limitation of the study is well examined but also the novelty of this modeling approach. The authors respond to every comment/suggestion made by reviewers and the paper is now improved.

Minor comment:

Line 32: "destroying terrestrial and aquatic ecosystems" please replace "destroying" with polluting or damaging

Reviewer #3:

Remarks to the Author:

The revised version looks better. But the rationale hidden behind the study is still questionable. The underlined science is very weak. For example, the time series estimations in both Figures 1 and 7 across several decades were presented by exact numbers with no any error bars to reflect uncertainty. For instance, the generation rates of waste across different countries could be very different. Simply having the issue can show the problem of lacking rigor of this study. The conclusion is neither helpful for decision/policy makers nor useful for engineers to retrieve anything for practical use. The only things left is a few global maps making the paper look a global study with no solid fecundation. Besides, the authors created a term "Circular Waste Management System" with no clear definition. In waste management community, no one used this term before in literature in the past 40 years except a previous paper form the same author group. The use of "Circular Waste Management Strategies" might be a lot more better. Leaking in the title is not appropriate as well. We normally use "illegal dumping". There are some writing issues still in English in this paper.

In conclusion, this paper is not mature enough for publication by the esteemed journal Nature - communication.

Reviewer #4:

Remarks to the Author:

1. What are the noteworthy results?

The paper is an exercise to estimate the leakage of MSW in aquatic environments at global and regional scales estimating the time evolution. The main objective seems to be the analysis of the circular economy influence on the leakage. A better explanation of model and parameters seems to be necessary in order to understand the influence of the circular economy in the results.

2. Will the work be of significance to the field and related fields?

The main conclusion is: "The leakage of MSW into aquatic environments is highly dependent on population size, populated area ((urban-rural), physical environment, waste generation and level of waste management" But this conclusion can be established from the beginning. The influence of the level of circular economy is not clearly shown in the conclusions.

3. How does it compare to the established literature? If the work is not original, please provide relevant references.

The paper follows the methodology of a previous paper (ref 1. Gomez Sanabria et als Nature Communications 2022), introducing some recent additional contributions of references related to water pollution and waste management (2-....).

4. Does the work support the conclusions and claims, or is additional evidence needed?

A better explanation of model and parameters seems to be necessary to see where the circular economy is introduced in the model

5. Are there any flaws in the data analysis, interpretation and conclusions? Do these prohibit publication or require revision?

A sensitivity and uncertainty analysis of the main parameters could clarify the meaning of the complex model and the circular economy in the system.

6. Is the methodology sound? Does the work meet the expected standards in your field?

Yes

6. Is there enough detail provided in the methods for the work to be reproduced?

Additional material referred to the mathematical model and parameters including sensitivity could help to understand better the results in the context of a circular economy discussion.

Comments to the author	Response from the author
Reviewer #1	
The revised manuscript is better structured with further explanations of results/ discussions related to other studies compared to the initial submission. The additional data and info provided in the SI materials were required to better understand the modeling process, results, and its shortcomings. The limitation of the study is well examined but also the novelty of this modeling approach. The authors respond to every comment/suggestion made by reviewers and the paper is now improved.	Thank you for this comment. We are happy to hear that we have addressed your comments correctly.
Line 32: “destroying terrestrial and aquatic ecosystems” please replace “destroying” with polluting or damaging	Thanks for your comment. We have replaced the word. Now it reads: “damaging terrestrial and aquatic ecosystems”.
Reviewer #2	
The revised version looks better. But the rationale hidden behind the study is still questionable. The underlined science is very weak. For example, the time series estimations in both Figures 1 and 7 across several decades were presented by exact numbers with no any error bars to reflect uncertainty. For instance, the generation rates of waste across different countries could be very different. Simply having the issue can show the problem of lacking rigor of this study.	Thanks for pointing this out. The main graphs depicting MSW generation and scattered waste (Fig 1) now include the (95% CI). We kept Fig 7 as it was in the main manuscript as it allows to rapidly compare the different scenarios and therefore helps to understand the results. We have added the CI for the figure in the Supplementary Data by GAINS country/region. Estimates of CI for the MSW generation model can be seen in the Table S6 in the Supplementary material. Estimates (including the 95% CI) of MSW generation and leakage by GAINS (country/region) are now included in the Supplementary Data (Excel file).
The conclusion is neither helpful for decision/policy makers nor useful for engineers to retrieve anything for practical use. The only things left is a few global maps making the paper look a global study with no solid fecundation	Thanks for pointing this out. We have worked on explaining the findings in a better way. We have also added one of the most important aspects in the abstract and conclusions which is the fact that waste collection rates are the main factor influencing waste leakage. This means, that even in a scenario in which waste is reduce but collection rates stagnate, the world will continue facing leakage of waste in terrestrial and aquatic environments.

	Our study also identifies the regions that need to be prioritized to reduce waste leakage into aquatic environments: “China, South Asia, Africa, and India account for the majority (~70%) of the potential leakage of MSW into aquatic environments. China is the region with the highest population residing within 1 km of rivers, South Asia is the region with the highest population residing within 1 km of coastal areas, and Africa with the highest population living close to lakes (up to 1 km). It is crucial to act accordingly in these regions to diminish the future potential MSW leakage associated with the lack of management systems. China and India have also been found as target regions to diminish leakage of plastic waste in the environment” Additionally, we highlight the difficulty of comparing different studies and identify the need for an improved reporting/standardized methods. Not just to identify areas in need of action but more importantly to track progress on it. Table S3, S4 and S5 show the aquatic systems that are most affected aquatic by waste leakage. In that sense, it provides information to decision and or policy makers about areas that should be prioritized. Further studies targeting individual areas can then complement this to guide direct interventions tackling the pollution problem.
Besides, the authors created a term "Circular Waste Management System" with no clear definition. In waste management community, no one used this term before in literature in the past 40 years except a previous paper from the same author group. The use of "Circular Waste Management Strategies" might be a lot more better.	Thank you for the suggestion. We fully respect the view expressed by the reviewer. Our study may be of interest to individuals from a multitude of fields and we could not determine that there is one single definition that suits all purposes. Looking at the overall use of the terms “Circular Waste Management Systems” or “Circular Waste Management Strategies” we find that the latter is used to the same extent, if not more, than the former even when discounting previously published literature by the authors of this study. Therefore, we have decided to stick with the term “Systems” as we are convinced it reflects more accurately the contents of our study. However, we acknowledge that more than one term could be used synonymously in this context. In the abstract, it is clarified that the implementation of a circular waste management system is considered a subset of potential mitigation strategies “we show the need for the adoption of active mitigation strategies, in particular circular waste management systems, that could stop waste from entering the aquatic ecosystems in the first place”

Leaking in the title is not appropriate as well. We normally use "illegal dumping". There are some writing issues still in English in this paper.	We have decided to keep the term waste leakage as the scientific and political community refers to plastic leakage following the definition: (Plastic) leakage refers to plastics (waste) that enter terrestrial and aquatic environments (https://www.oecd-ilibrary.org/sites/e427e93c-en/index.html?itemId=/content/component/e427e93c-en, https://www.igi-global.com/dictionary/sources-and-pathways-of-marine-litter/110246). In our case, it refers to waste leakage that enter the terrestrial and aquatic environments.
In conclusion, this paper is not mature enough for publication by the esteemed journal Nature - communication.	We would like to thank you for your review. It definitely has helped us to significantly improve the paper.
Reviewer #3	
1. What are the noteworthy results? The paper is an exercise to estimate the leakage of MSW in aquatic environments at global and regional scales estimating the time evolution. The main objective seems to be the analysis of the circular economy influence on the leakage. A better explanation of model and parameters seems to be necessary in order to understand the influence of the circular economy in the results.	Thanks for pointing this out. We believe that indeed a better explanation of the model and parameters was needed. Thanks a lot for noticing this. We have added a section on the methods section explaining the parameters in the model and how the model represents circular waste management systems as follows: Methods The Greenhouse Gas-Air Pollution Interactions and Synergies (GAINS)⁴⁹ (http://gains.iiasa.ac.at/) model from the International Institute of Applied Systems Analysis is used to carry out this assessment, especially the municipal solid waste module^{2,45,50}. The GAINS model is an integrated assessment model that provides an authoritative framework for assessing strategies that reduce emissions of multiple air pollutants and GHG at least cost, and minimize their negative effects on human health, ecosystems, and climate change. The GAINS model that has global coverage with a geographic representation of 180 country/regions (Table S4 lists the countries/regions in GAINS) with multitemporal resolution at five years intervals. The methodology to estimate potential leakage of MSW and identification of hotspots involves the following steps: 1. Model parametrization: The municipal solid waste module in GAINS integrates socio-economic variables which are exogenous to the model (i.e., population, urbanization, GDP per capita), waste generation quantities, waste composition, waste control measures (i.e., waste treatment) and emission factors to quantify waste flows and to estimate air pollutants and GHG emissions. The municipal solid waste module in the model

	differentiates between urban and rural areas within a country/region and types of waste management by stream. This allows to capture national and regional disparities to a certain extent. The model links waste generation and waste composition to the type of treatment by applying the matrix presented in Table 1 (please see table below). The model constraints the treatments when no applicable to a specific waste fraction. The applicability of the waste control measures by type of waste should always sum up to 100%. The model allows to represent actions and political interventions such as reduction of waste by intervening the activity data (e.g., waste generation and composition), represent variations/improvements in waste management systems by modifying the applicability rate of the waste control measures, and quantify the circularity of the waste systems by indicators such as landfill diversion rates, recycling rates by stream, anaerobic digestion and composting rates, carbon flows and energy generation. By contrasting baseline with mitigation scenarios, the model provides an analytical information to support the analysis of environmental, economic, and social impacts. In this context, the municipal solid waste module in GAINS offers the opportunity to analyze the co-benefits of adopting circular waste management systems on GHG and air pollutants emissions and as a result of this study leakage of waste into aquatic environments.
2. Will the work be of significance to the field and related fields? The main conclusion is: "The leakage of MSW into aquatic environments is highly dependent on population size, populated area ((urban-rural), physical environment, waste generation and level of waste management" But this conclusion can be established from the beginning. The influence of the level of circular economy is not clearly shown in the conclusions.	Thanks for pointing this out. We have work on explaining in a better way how we have adopted the circular economy concept (see point 1) and have improved the conclusions. We have also added one of the most important aspects in the abstract and conclusions which is the fact that waste collection rates are the main factor influencing waste leakage: "However, even in a scenario representing a sustainable world, in which technical, social, and financial barriers are overcome and public awareness and participation to rapidly increase waste collection rates, reduce, reuse and recycling waste exists, it would be impossible to entirely eliminate waste leakage before 2030, failing to meet the waste-related Sustainable Development Goals." This means, that even in a scenario in which waste is reduced but collection rates stagnate, the world will continue facing leakage of waste in terrestrial and aquatic environments In addition, our study also identifies the regions that need to be prioritized to reduce waste leakage into aquatic environments: "China, South Asia, Africa, and India account for the majority (~70%) of the potential leakage of MSW into aquatic environments.

	China is the region with the highest population residing within 1 km of rivers, South Asia is the region with the highest population residing within 1 km of coastal areas, and Africa with the highest population living close to lakes (up to 1 km). It is crucial to act accordingly in these regions to diminish the future potential MSW leakage associated with the lack of management systems. China and India have also been found as target regions to diminish leakage of plastic waste in the environment” Table S3, S4 and S5 show the aquatic systems that are most affected by waste leakage. In that sense, it provides information to decision and or policy makers about areas that should be prioritized.
3. How does it compare to the established literature? If the work is not original, please provide relevant references. The paper follows the methodology of a previous paper (ref 1. Gomez Sanabria et al Nature Communications 2022), introducing some recent additional contributions of references related to water pollution and waste management (2-....).	Thanks for your review. Certainly, the methodology of this study relies on our previous studies. Furthermore, in order to be able to assess waste leakage into aquatic environments, we have consulted more recent studies that had been cited thorough out the paper and its supplementary information. Furthermore, this study expands the previous research by applying its concepts to additional analysis, for example spatial aspects accounting for differences in urbanized or rural areas.
4. Does the work support the conclusions and claims, or is additional evidence needed? A better explanation of model and parameters seems to be necessary to see where the circular economy is introduced in the model	Thanks for this comment. We believe that indeed a better explanation of the model and parameters was needed. Thanks a lot for noticing this. Please refer to the response of your first comment.
5. Are there any flaws in the data analysis, interpretation and conclusions? Do these prohibit publication or require revision? A sensitivity and uncertainty analysis of the main parameters could clarify the meaning of the complex model and the circular economy in the system.	Thanks for pointing this out. The main graphs depicting MSW generation and scattered waste (Fig 1) now include the (95% CI). We kept Fig 7 as it was in the main manuscript as it allows to rapidly compare the different scenarios and therefore helps to understand the results. We have added the CI for the figure in the Supplementary Data by GAINS country/region. Estimates of CI for the MSW generation model can be seen in the Table S6 in the Supplementary material. Estimates (including the 95% CI) of MSW generation and leakage by GAINS (country/region) are now included in the Supplementary Data (Excel file).

6. Is there enough detail provided in the methods for the work to be reproduced? Additional material referred to the mathematical model and parameters including sensitivity could help to understand better the results in the context of a circular economy discussion.	We have included the explanation of the model parameters (see reply comment 1) to make the methodology clearer. The circular economy is included in the context of reducing waste generation (e.g., in the SSP1 – sustainability scenario), diverting waste from landfills by increasing recycling rates of materials and increasing anaerobic digestion/composting of food waste. It also includes the upgrade of landfills to sanitary landfills with gas recovery and energy use. The adoption of the different strategies varies according to the SSP narratives. Description of the narratives can be found in section S6 of the supplementary material. We have also ran a Montecarlo simulation and performed a sensitivity analysis (100 samples) using the packages sensitivity, randtoolbox and lhs in R studio. In our study we use the coefficient of variation to identify input parameters with higher relative variability. The results indicate that uncollected waste is the variable with a potential more significant impact on waste leakage (Supplementary Information S9 shows the average CV for the simulations).
--	--

Table 1. GAINS municipal solid waste matrix

Solid waste management technology	Municipal solid waste							
	Food	Glass	Metal	Other	Paper	Plastic	Textile	Wood
Open burned	X			X	X	X	X	X
Scattered and/or disposed to water-courses	X	X	X	X	X	X	X	X
Unmanaged solid waste disposal site - low humidity - < 5m deep	X			X	X		X	X
Unmanaged solid waste disposal site - high humidity - > 5m deep	X			X	X		X	X
Compacted landfill	X	X	X	X	X	X	X	X
Covered landfill	X			X	X		X	X
Landfill gas recovery and flaring	X			X	X		X	X
Landfill gas recovery and used	X			X	X		X	X

Incineration (poor air quality controls)	X			X	X	X	X	X
Incineration (high quality air pollution controls - energy recovery)	X			X	X	X	X	X
Anaerobic digestion	X							
Composting	X							
Recycling		X	X		X	X	X	X

Source: (Gómez-Sanabria et al., 2018)⁴⁵

Reviewers' Comments:

Reviewer #2:

Remarks to the Author:

This revised version of the manuscript added further clarifications to the methods section plus the uncertainty analysis with additional info on SI files.

In my opinion, this paper could be accepted for publication.

Reviewer #3:

None

Reviewer #4:

Remarks to the Author:

tTaking into account the uncertainties in the evolution of the parameters the paper shows a simplified picture of the global transfer of municipal wastes to water based on the GAINS model and software.

Following the definition of the text:

"A circular waste management system is defined here as a system with successful implementation of MSW reduction policies by reducing food and plastic waste generation, maximum technically feasible recycling rates of all MSW streams, and once recycling capacity is exhausted, incineration of refuse MSW with energy recovery. Furthermore anaerobic digestion is implemented to treat food and garden waste, high diversion of MSW from landfills and upgrading of dumpsites⁴⁵"

According to the definition of CWM in the paper the rates of reduction, reuse and recycling of MSW are going to be the parameters identifying the influence of CWM in the system.

I can understand that these technical parameters are going to be the responsible of a Zero Municipal Wastes policy.

On the other side I consider that the materials flow methods allow a LCA approach from the cradle to the grave in the system, where recycling is the technical flow to be optimized and human behaviour the other main influence.

The accumulation of Wastes in Water has to be connected with the inlet and outlet and decomposition rates.

Comments to the author	Response from the author
A section clarifying the limitations of the analysis must be introduced before publication, particularly with regards to potential limitations in linking the analysis here with real world conditions.	Thanks for your comment. Certainly, there are some limitations in applying the research to the real world. We have added the following paragraph in the uncertainties and limitations section: While the databases and information we use represent the best available to our knowledge on a global level, we are aware that this analysis cannot fully capture the complexity of real-world scenarios. The data and methods applied each come with their own limitations, such as data quality and availability or characteristics of the physical world which could not be modelled. Aspects such as future variations in environmental conditions, geopolitical unrest or unforeseen events could limit the applicability of the results additionally. This underlines the importance of developing and maintaining Monitoring, Reporting and Verification frameworks to validate and adapt the analysis, aligning it further with the development of the world.
Reviewer #2	
This revised version of the manuscript added further clarifications to the methods section plus the uncertainty analysis with additional info on SI files. In my opinion, this paper could be accepted for publication.	Thank you for this comment. We are happy to hear that we have addressed your comments correctly.
Reviewer #4	
tTaking into account the uncertainties in the evolution of the parameters the paper shows a simplified picture of the global transfer of municipal wastes to water based on the GAINS model and software. Following the definition of the text: "A circular waste management system is defined here as a system with successful implementation of MSW reduction policies by reducing food and plastic waste generation, maximum technically feasible recycling rates of all MSW streams, and once recycling capacity is exhausted, incineration of refuse MSW	Thanks for your comment. We have not analyzed the decomposition rates of the different waste streams as this is out of our scope. We recognize that this could be an interesting enhancement for a potential future analysis. We attempt to identify the “hotspots” where MSW is at high risk of entering water courses in a given point in time in order to call for action to avoid the leakage of waste in the first place. Therefore, the decomposition and accumulation of waste over time are beyond our analysis. In addition, analyzing the decomposition of each of the waste streams requires a separate study and the involvement of scientists from different disciplines such as biologist, biochemists, oceanographers, among others.

with energy recovery. Furthermore anaerobic digestion is implemented to treat food and garden waste, high diversion of MSW from landfills and upgrading of dumpsites⁴⁵" According to the definition of CWM in the paper the rates of reduction, reuse and recycling of MSW are going to be the parameters identifying the influence of CWM in the system. I can understand that these technical parameters are going to be the responsible of a Zero Municipal Wastes policy. On the other side I consider that the materials flow methods allow a LCA approach from the cradle to the grave in the system, where recycling is the technical flow to be optimized and human behaviour the other main influence. The accumulation of Wastes in Water has to be connected with the inlet and outlet and decomposition rates.

For example, several studies look at the decomposition of plastics in the marine environment (<https://www.sciencedirect.com/science/article/pii/S0025326X19308896>, <https://www.nature.com/articles/s41467-023-44368-8>, <https://www.sciencedirect.com/science/article/pii/S1878535222005780>)